# HelpSteer2-Preference: Complementing Ratings with Preferences[*]

**Zhilin Wang**[1]     **Alexander Bukharin**[1,2]     **Olivier Delalleau**[1]     **Daniel Egert**[1]

**Gerald Shen**[1]     **Jiaqi Zeng**[1]     **Oleksii Kuchaiev**[1]     **Yi Dong**[1]

{zhilinw, yidong}@nvidia.com

## Abstract

Reward models are critical for aligning models to follow instructions, and are typically trained following one of two popular paradigms: Bradley-Terry style or Regression style. However, there is a lack of evidence that either approach is better than the other, when adequately matched for data. This is primarily because these approaches require data collected in different (but incompatible) formats, meaning that adequately matched data is not available in existing public datasets. To tackle this problem, we release preference annotations (designed for Bradley-Terry training) to complement existing ratings (designed for Regression style training) in the HelpSteer2 dataset. To improve data interpretability, preference annotations are accompanied with human-written justifications. Using this data, we conduct the first head-to-head comparison of Bradley-Terry and Regression models when adequately matched for data. Based on insights derived from such a comparison, we propose a novel approach to combine Bradley-Terry and Regression reward modeling. A Llama-3.1-70B-Instruct model tuned with this approach scores 94.1 on RewardBench, emerging top of more than 140 reward models as of 1 Oct 2024. This reward model can then be used with REINFORCE algorithm (RLHF) to align an Instruct model to reach 85.0 on Arena Hard, which is No. 1 as of 1 Oct 2024.

🤗 **Dataset (CC-BY-4.0-License):** huggingface.co/datasets/nvidia/HelpSteer2
🤗 **Reward Model:** huggingface.co/nvidia/Llama-3.1-Nemotron-70B-Reward
🤗 **Instruct Model:** huggingface.co/nvidia/Llama-3.1-Nemotron-70B-Instruct

## 1 Introduction

First featured in the Reinforcement Learning from Human Feedback (RLHF) pipeline for aligning language models to follow instructions (Bai et al., 2022; Ouyang et al., 2022), Reward Models are still prominently featured as a critical part for aligning frontier open-recipe models (Dubey et al., 2024; Nvidia et al., 2024). The role of Reward Models lies in assigning high scores to responses that follow instructions in a helpful and safe manner while assigning low scores to those that do not. This in turn guides language models to generate more responses that give high scores, which make them more helpful and safer (Dong et al., 2024; Lambert et al., 2024).

While training a reward model that can accurately separate good responses from bad ones is a consensus goal, there is less agreement on the best path to get there. On one side are the traditional Bradley-Terry Style Reward Models first introduced by Bai et al. (2022) and Ouyang et al. (2022) which seek to maximize the gap in reward between chosen and rejected responses to the same prompt. On the other side are Regression style Reward Models introduced by Wang et al. (2024e) and Wang et al. (2024a) and have lately been used to train some of the top models on RewardBench (AllenAI, 2024) such as ArmoRM (Wang et al., 2024b) and Nemotron-4-340B-Reward (Wang et al., 2024d). Regression Reward Models train the model to predict the score (often Likert-5) for a response to a

---

[*][1]NVIDIA, [2]Georgia Tech, work done during internship at NVIDIA

particular prompt. For many researchers and practitioners, deciding which style of Reward Models to adopt is challenging due to the absence of empirical studies that directly compare them when appropriately matched for data, which means:

1. **Identical set of prompts and responses** This mitigates the confounding factor of prompts and responses influencing reward model performance, which plays a role outside the data collection approach.

2. **Collected for Purpose** Data used to train each type of Reward Model should be collected in the format they will be used. While there are heuristics to convert Regression-style ordinal annotations over to preference annotations (using the difference between scores of individual responses), our early experiments found them to be comparatively lackluster for training Bradley-Terry models. We hypothesize that Regression-style data collection (where responses are rated individually) gives annotators a different set of expectations for comparative ranking of responses. For instance, in the Likert-5 scale used by HelpSteer2 (Wang et al., 2024d), both responses can be given a helpfulness 3 but one might still be preferable to the other. Post-hoc heuristics used to convert such Regression-style data to preference rankings are unable to account for such nuances in preferences.

3. **High Quality** Following the adage 'Garbage in Garbage out', the role that the data format (regression-style vs. preferences) plays may not reliably show when the quality of data collection is the bottleneck, as the signal-to-noise ratio will be too low for the model to learn anything useful. Shen et al. (2024) found that the Llama-2-7B-Chat model performs essentially at a random level (i.e. <55% for a binary choice task) on RewardBench Chat-Hard category (Lambert et al., 2024) when trained on Ultrafeedback (Cui et al., 2023) or HH-RLHF (Bai et al., 2022), which is attributed to noise in the ground truth labels. Similarly, Wang et al. (2024d) found that Llama-3-70B models trained on HH-RLHF (Bai et al., 2022), Open Assistant (Köpf et al., 2023) or HelpSteer (Wang et al., 2024e) do not surpass 60% on the same evaluation. In contrast, Llama-3-70B trained on HelpSteer2 (Wang et al., 2024d) performs above 80%. Without meeting a high bar on data quality, it may not be possible to discern the advantages of a particular data annotation methodology over another.

To the best of our knowledge, no one has thus far publicly released data that is adequately matched for both approaches (see detailed Related Works in Appendix A). In this work, we release preference annotations that were collected alongside the Likert-5 ratings from HelpSteer2 (Wang et al., 2024d), a high quality dataset used to train some of top models on RewardBench (*e.g.* Nemotron-4-340B-Reward). [1] We show that Bradley-Terry Models can be effectively trained with such preference annotations, and also investigate leveraging preference justifications where annotators indicate why they preferred one response over another.

**Our key contributions are:**

1. We open-source (CC-BY-4.0) a high-quality preference modeling dataset containing preference directions, strengths, and justifications. To the best of our knowledge, this is the first open-source release of a general-domain preference dataset containing *human-written* preference justifications.

2. Using this data, we perform a head-to-head comparison of Bradley-Terry style and Regression style Reward Models, alongside reward models that can make use of preference justifications.

3. From insights derived from the above comparison, we propose a novel approach to combine Bradley-Terry and Regression Reward Models, which can be used to train a reward model that scores 94.1 on RewardBench, which is the best performing model as of 1 Oct 2024. This reward model can then be used with the REINFORCE algorithm (RLHF) to align a model to reach 85.0 on Arena Hard.

## 2 DATASET

**Data Collection** For each task, annotators are provided a prompt and two responses. They first annotate each response on a Likert-5 scale along several dimensions (helpfulness, correctness and coherence, complexity and verbosity), as detailed in Wang et al. (2024d). Then, they choose between 7 preference options, each associated with a preference score as well as a justification for their preference:

---

[1]This data has not been released previously because we did not have sufficient resources to conduct experiments to demonstrate the value of preference annotations.

-3. Response 1 is much better than Response 2

-2. Response 1 is better than Response 2

-1. Response 1 is slightly better than Response 2

1. Response 2 is slightly better than Response 1

2. Response 2 is better than Response 1

3. Response 2 is much better than Response 1

-100. Neither response is valid

The last option ("Neither response is valid") is to be used when both responses are so bad that trying to identify a winner is pointless. Inspired by Bai et al. (2022) and Touvron et al. (2023), we force the annotator to make a preference choice between the two responses (i.e. no option for Both responses are equally good), except for "Neither response is valid". Our motivation was to reduce sitting-on-the-fence behavior and encourage annotators to look closer at differences (even if they *appear* to be minor), in order to provide more robust preference information. The preference ranking guidelines and examples provided to annotators are in the Appendix B.

The vendor (Scale AI) was asked to distribute each task to 3-5 annotators to independently label preference among two responses for every prompt. For a small proportion of samples ($<10\%$), there are fewer than three useful annotations in the resulting dataset. This is due to annotators skipping some tasks as they were unable to rate them effectively or indicating that both responses were invalid (i.e. rated as -100, which were excluded).

**Data Pre-processing**  In line with HelpSteer2, we identify the three most similar preference annotations per task (to avoid interference by outliers), take their mean, and round it to the closest integer to give the overall preference. Furthermore, we filter out 10% of tasks, for which the spread of annotations of the three most similar annotations was more than two. For instance, a task with individual preferences of [-3, -1, 2, 3] will have the most similar annotations of [-1, 2, 3] with a spread of 4, and thus excluded. This is done to avoid training on tasks for which the ground-truth preference cannot be confidently estimated among human annotators. 22% of the samples have an overall preference of 0 in situations where annotators disagree (e.g. [-1, -1, 1]) and these samples are also excluded because a near-zero average indicates low-confidence preferences, which may introduce noise during Reward Model training. Overall, we have 7,118 preference pairs with 6,766 pairs in the training set and 352 pairs in the validation set.

To compare the inter-rater reliability of our collected data (compared to HelpSteer2), we follow Wang et al. (2024d), to use quadratic-weighted Cohen's $\kappa$ (Artstein & Poesio, 2008) as a measure of inter-rater agreement. The quadratic weighted version of Cohen's $\kappa$ (Scikit-Learn, 2024) can also penalize larger disagreements (e.g. -3 and +3) much more heavily compared to smaller disagreements (e.g. -1 and +1). The agreement of the raw preferences is 0.489 (moderate), suggesting that having annotators agree on the direction and strength of preferences is challenging. Our pre-processing steps (i.e. using the three most similar annotations per task and removing samples with a preference spread of more than two) increase Cohen's $\kappa$ to 0.843 (strong), suggesting that weeding out outlier annotations is an effective way to improve agreement. Finally, excluding samples with an overall preference of 0 further increases Cohen's $\kappa$ to 0.878 (strong). The observed final agreement is stronger than HelpSteer2 helpfulness (with 0.791), suggesting good reliability of our collected data. We detail how we pre-process preference justifications in Appendix C.

**Data Analysis**  We analyze the dataset including samples for which the overall preference is zero, as this can provide insights below, even though such samples are not used for training subsequently. As shown in Fig. 1, the distribution of preferences is concentrated at the center (A=B or 0) and reduces gradually away from the center ($\mu = 0.0649$, $\sigma = 1.72$), with a slight bias to preferring Response 2 over Response 1. This means that Response 2 is preferred in 41.6% of tasks while Response 1 is preferred in only 36.5%. This bias is especially high for the "slightly better" setting where Response 2 is slightly preferred 18.5% vs. Response 1 14.8%. Such a bias is similar in extent to the difference between the helpfulness score of Response 1 and Response 2 from the original HelpSteer2 dataset (Wang et al., 2024d): 39.3% of tasks gave Response 2 a higher helpfulness score while only 33.9% gave Response 1 a higher helpfulness score.

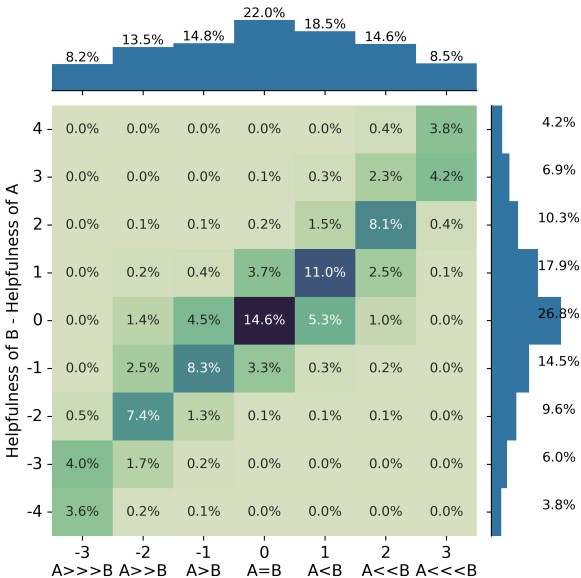

Figure 1: Distribution of preferences between responses in HelpSteer2-Preference against the difference in helpfulness scores between them from HelpSteer2. For clarity, A refers to Response 1 while B refers to Response 2. >>> means much better, >> means better, > means slightly better and = means as good as.

There are some possible reasons for such a position bias. First, the sampling of responses between Response 1 and 2 might not have been even, with Response 2 having been dis-proportionally generated by stronger models. Second, annotators could also have a positional bias, in which the order of responses shown to them can influence their judgment. In this case, the response they see later could be making a stronger impression on them, leading annotators to dis-proportionally prefer them due to a recency bias (Phillips-Wren et al., 2019). While it is important to note the existence of this position bias, it is relatively small in extent (5.1% difference in preference), which minimally affects their usefulness compared to LLM-as-a-judge settings (with GPT-4 and Claude) where the position bias could be between 25% and 75% (Zheng et al., 2023).

Fig. 1 also shows that the distribution in preference between the two responses is highly correlated with the difference in the responses' helpfulness score from HelpSteer2 (Pearson's R = 0.9024). When two responses have the same helpfulness score, they most likely are as good as each other but in slightly less than half the cases (45.5%), annotators still show a *consistent* preference for one response. This supports our initial hypothesis that preference ranking can produce a more fine-grained separation between two responses even if they share the same Likert-5 helpfulness score. Slight preferences are most commonly (58.0%) associated with a difference in helpfulness scores of one while strong preference preferences are frequently (93.4%) associated with a helpfulness difference of 3 or 4. Interestingly, there are also substantial samples (7.4%) that show no preference despite having an absolute difference of one or more. This is likely due to annotator disagreements, which are averaged to obtain the final overall preference. Given the constraints on space, we defer an extended analysis of preference justifications (both structural and semantic) to Appendix D.

# 3 REWARD MODEL

## 3.1 EVALUATION

Following Dong et al. (2024); Wang et al. (2024d); Yuan et al. (2024), we perform evaluation using RewardBench (Lambert et al., 2024), a trusted reward modeling benchmark with over 140 models on the public leaderboard (AllenAI, 2024). RewardBench comprises 2985 diverse tasks across 4 categories - Chat, Chat-Hard, Safety, and Reasoning - and 23 sub-categories), which minimizes the likelihood of over-fitting to particular tasks. Each task consists of a prompt, a chosen response, and

a rejected response. The Chat category involves comparing a bad and a good model response in general-domain conversation settings while Chat-Hard does the same for a good model response compared to a great model response. The Safety category measures the preference between a refusal response and a compliance response to an unsafe user request. The Reasoning category assesses the preference between correct and incorrect responses related to math and coding prompts. The accuracy for each category is calculated based on the proportion of tasks it gets correct (i.e. prefer chosen over rejected), except for the Reasoning category, which up-weighs math samples to ensure math and coding prompts contribute equally. Overall accuracy is calculated using the mean of the four category scores.

## 3.2 TRAINING

**SteerLM Regression**  Following Wang et al. (2024d), we train SteerLM Regression Reward Models consisting of a base model and a linear layer projecting the final layer dense representation of the end-of-response token into five scalars, one for each HelpSteer2 attribute. Such models are optimized using a MSE loss function, which seeks to minimize the squared error between the predicted attribute value and the ground truth attribute value. In addition, we train a separate model only on the Helpfulness attribute.

**Bradley-Terry**  Following Ouyang et al. (2022); Bai et al. (2022), we train Bradley-Terry style Reward Models, consisting of a base model and a linear layer projecting the final layer dense representation of the end-of-response token into a scalar reward. Models are trained to maximize the directional distance between the reward for the chosen response ($y_c$) and the rejected response ($y_r$), as in Eq. 1, thereafter referred to as Regular BT.

$$\mathcal{L}_{\mathcal{BT}} = -\log\left(\sigma(r_\theta(x, y_c) - r_\theta(x, y_r))\right) \tag{1}$$

Given that HelpSteer2-Preference contains not only the direction of preference between two responses but the magnitude ($m$) of this preference (1 - slightly better, 2 - better, 3 - much better), we also experiment with a Bradley-Terry with Margin loss introduced by Touvron et al. (2023) in Eq. 2, thereafter referred to as Margin BT.

$$\mathcal{L}_{\mathcal{MBT}} = -\log\left(\sigma(r_\theta(x, y_c) - r_\theta(x, y_r) - m)\right) \tag{2}$$

We also introduce a new loss function named Scaled Bradley-Terry in Eq. 3, hereafter referred to as Scaled BT. Similar to Margin BT, its motivation lies in utilizing the preference magnitude information. However, we use the margin term outside of the log-sigmoid function rather than inside. From the perspective of data utilization, this can be viewed as a repeated sampling of response pairs for which the preference magnitude is higher. From the perspective of model training, this can be seen as requiring models to learn more (i.e. larger update) from responses that we know to be greatly different from each other (more in Appendix J). Unlike Margin BT, Scaled BT does not assume that the distance between the chosen and rejected rewards needs to be as least as large as the margin term.

$$\mathcal{L}_{\mathcal{SBT}} = -m\log\left(\sigma(r_\theta(x, y_c) - r_\theta(x, y_r))\right) \tag{3}$$

Finally, we also train BT models initialized on the Helpfulness-Only SteerLM Regression Model. The Regression model is trained to predict helpfulness between 0 and 4, which can potentially initialize the model better than the base model, which otherwise has high loss at the start of training.

**Pairwise Justifier**  To explore training reward models using preference justifications rather than preference scores, we train Pairwise Justifier reward models similar to how proprietary models such as GPT-4-Turbo are used for LLM-as-a-Judge in AlpacaEval (Taori et al., 2023) and Arena-Hard (Li et al., 2024). In these settings, the LLM is prompted to generate a detailed comparison of the two responses before finally generating a statement such as "Response 1 is better than Response 2". HelpSteer2-Preference provides a unique opportunity for examining whether preference justifications can be used to train more accurate reward models (compared to using preference score) when training data is kept the same. To train such models, we seek to generate the preference

justification conditioned on `{prompt} @Response 1:\n{response_1}\n@Response 2:\n{response_2}\nBetween @Response 1 and @Response 2, which is better?`. This model is then optimized using a Cross-Entropy loss as typically used for supervised fine-tuning of language models. In this setting, we format the preference justification by concatenating preference elaboration followed by preference statement, which is always in the format `@Response 1/2 ... better`. This allows automatic extraction of the better response for evaluation. We also experiment with *i.* including up to three preference justifications (*i.e.* All Justifications set) instead of one per task and *ii.* augmenting the dataset by flipping Response 1 with Response 2 and correspondingly updating the preference label (which also minimizes position bias) and *iii.* ablations to understand the role of each component.

**Training Details**    In our experiments, we use Llama-3.1-70B-Instruct (Dubey et al., 2024) as the base model. Our initial exploration of training reward models showed that Llama-3.1-70B-Instruct performs better as an initial model than Llama-3.1-70B as well as Nemotron 4 340B (Nvidia et al., 2024) and Llama-3-70B (Meta AI, 2024), as used by Wang et al. (2024d). Hyper-parameters for training and the search range for them are in Appendix E.

# 4    REWARD MODEL RESULTS

| Model Type | Model | Overall | Chat | Chat-Hard | Safety | Reasoning | LR |
|---|---|---|---|---|---|---|---|
| | | | | **RewardBench** | | | **Hyperparams** |
| *SteerLM Regression* | HelpSteer Attributes | 92.4 | 95.0 | 85.5 | 94.0 | 95.1 | 1e-6 |
| | Helpfulness Only | 93.0 | 97.2 | 84.2 | 94.6 | 95.8 | 2e-6 |
| *Bradley-Terry* | Regular | 91.5 | 97.5 | 80.3 | 90.5 | 97.9 | 3e-6 |
| (from scratch) | Margin | 91.5 | 98.0 | 78.5 | 94.6 | 94.8 | 2e-6 |
| | Scaled | 92.7 | 97.8 | 83.5 | 93.2 | 96.0 | 2e-6 |
| *Bradley-Terry* | Regular | 92.7 | **98.9** | 82.9 | 93.7 | 95.4 | 1e-6 |
| (init. with Helpfulness- | Margin | 93.0 | 98.3 | 83.8 | 94.0 | 95.8 | 1e-6 |
| only Regression Model) | Scaled | 93.7 | 98.0 | 85.7 | 94.3 | 96.7 | 1e-6 |
| | Scaled + ExPO | **94.1** | 97.5 | 85.7 | **95.1** | **98.1** | 1e-6 |
| *Pairwise Justifier* | Full Preference Justification | 88.1 | 94.4 | 82.2 | 90.9 | 84.9 | 3e-6 |
| | (- three justifications per task) | 86.1 | 96.1 | 71.7 | 89.5 | 87.2 | 2e-6 |
| | (- augment data by flipping labels) | 86.8 | 95.8 | 76.1 | 89.5 | 86.0 | 3e-6 |
| | (- Preference Elaborations) | 88.4 | 96.4 | 78.7 | 93.4 | 85.3 | 1e-6 |
| | (- Preference Justifications *i.e.* only Response 1 or 2 as label) | 90.0 | 96.1 | 80.0 | 93.1 | 90.9 | 3e-6 |
| *External Baselines* | Skywork-Reward-Gemma-2-27B | 93.8 | 95.8 | **91.4** | 91.9 | 96.1 | |
| | TextEval-Llama3.1-70B | 93.5 | 94.1 | 90.1 | 93.2 | 96.4 | |
| | Skywork-Critic-Llama-3.1-70B | 93.3 | 96.6 | 87.9 | 93.1 | 95.5 | |
| | SFR-LLaMa-3.1-70B-Judge-r | 92.7 | 96.9 | 84.8 | 91.6 | 97.6 | |
| | Nemotron-4-340B-Reward | 92.0 | 95.8 | 87.1 | 91.5 | 93.6 | |
| | Llama3-70B-SteerLM-RM | 88.8 | 91.3 | 80.3 | 92.8 | 90.6 | |
| | GPT-4o-2024-08-06 | 86.7 | 96.1 | 76.1 | 88.1 | 86.6 | |
| | Claude-3.5-Sonnet-20240620 | 84.2 | 96.4 | 74.0 | 81.6 | 84.7 | |
| | Meta-Llama-3.1-70B-Instruct | 84.0 | 97.2 | 70.2 | 82.8 | 86.0 | |

Table 1: Performance of Models on RewardBench. Higher is better for each category. All models are trained by us using Llama-3.1-70B-Instruct as a base model except External Baselines, the scores for which are taken from RewardBench leaderboard (AllenAI, 2024)

**SteerLM Regression**    As shown in Table 1, training a SteerLM Regression only on the Helpfulness attribute leads to slightly better performance on RewardBench overall (93.0 vs 92.4), relative to training on all five HelpSteer attributes as proposed by Wang et al. (2024d). While training on all five HelpSteer attributes can provide more insights to other dimensions of the response (correctness, coherence, complexity, verbosity), training with only the helpfulness attribute also makes the setup easier for training and inference. Specifically, there is no longer a concern that the five objectives/dimensions might conflict and the reward model produces a singular scalar reward without needing to derive one using a weighted sum of the five attribute values.

**Bradley-Terry (from scratch)**    Scaled BT performs much better than either Regular BT or Margin BT on RewardBench Overall at 92.7 vs. 91.5. This is likely because Scaled BT can most effectively use the preference magnitude information to guide model training.

**SteerLM Regression vs. Bradley-Terry**  To answer our initial question about whether SteerLM Regression or Bradley-Terry is better, we find that the optimal formulation of each variant (Helpfulness-Only and Scaled BT) performs just about as well as each other on RewardBench Overall (92.7 vs 93.0). This suggests that the format that the data is collected in and the model training approach does not matter too much. Instead, the most important consideration is that the modelling details can fully account for the information captured in the annotation. In the case of Bradley-Terry models, the magnitude of preference strength should be adequately modelled.

**Complementing SteerLM Regression with Bradley-Terry**  While SteerLM Regression and Bradley-Terry models are separately comparable, we show that they have a synergistic effect and result in a stronger reward model when combined. Specifically, when initialized with the Helpfulness-only Regression model, a Scaled Bradley-Terry can reach overall RewardBench of 93.7. This is likely a result of the HelpSteer2 and HelpSteer2-Preference datasets containing complementary information, as first indicated by Sec. 2. Conceptually, this synergy is similar to the two-stage approach in performing preference tuning (i.e. Proximal Policy Optimization or Direct Preference Optimization) after doing supervised-finetuning. In addition, we found ExPO (Zheng et al., 2024) to be a simple and effective way of extrapolating the delta weights to further improve the model. Specifically, we use the Helpfulness-Only SteerLM Regression model as the weak model and the Scaled BT model (initialized with the Helpfulness-Only SteerLM Regression model) as the strong model. We then did an extrapolation factor search between 1.1 and 2.0 at intervals of 0.1. Upon finding 1.6 to be optimal, we searched between 1.51 and 1.69 at intervals of 0.01. The final extrapolation factor was 1.52. Neither the Regular BT model nor the Margin BT model improved upon the Helpfulness-only Regression Model that they were initialized with and therefore we did not try ExPO on them. To better appreciate the contribution of certain design choices, we report ablation studies in Appendix K.

**Pairwise Justifier**  Compared to SteerLM Regression and Bradley-Terry models that can score each sample independently, Pairwise Justifier models which involve choosing a better response between two candidates generally perform worse, with an overall RewardBench score of 90.0 or lower. We suspect that this is because this task formulation is much harder (implicitly involving scoring Response 1, then scoring Response 2 and finally determining which score is higher). This is also supported by the observation that strong external baseline models using this Pairwise Justifier approach (*e.g.* gpt-4o-2024-08-06 and Meta-Llama-3.1-70B-Instruct) score 86.7 and 84.0, which are similar to our Pairwise Justifier models. On the other hand, SteerLM Regression and Bradley-Terry models both decompose this problem into simpler objectives that can be directly optimized towards.

As seen in Table 1, training a model on more than one preference justification per task is helpful (overall score increases from 86.1 to 88.1%), presumably because it provide a greater diversity of reasoning paths leading to the final preference choice. This increase is mostly contributed by gains on Chat-Hard (71.7 to 82.2%) while other categories do not change much. This is likely because our training data collection objective is most aligned with the Chat-Hard category (i.e. separating great general domain responses from good ones), meaning that extrapolating such reasoning to specialized-domains such as safety and reasoning (math/code) can be challenging. Data Augmentation through flipping labels is also helpful (1.3% increase in Overall), likely as it reduce the effects of position bias, which is present in our data, albeit to a small extent (5.1% difference in preference). Interestingly, training on just the Preference label (Response 1 or 2) does better than train on either the Preference Statement only (w/o Elaboration) or the Full Preference Justification. This is likely because the model can implicitly learn *why* a response might be better than the other in a way that's more effective than grounding the choice based on human-written explanations. Nonetheless, it is important to bear in mind that the current results are based on a small general-domain dataset comprising only 6.6k tasks, written in a free-form manner. Having a dataset which is different in domain, size or guidance to annotators (*e.g.* more structured) can lead to different conclusions. We leave further explorations to future work.

**Comparison with External Baselines**  Relative to the top performing external baseline (Skywork-Reward-Gemma-2-27B), our best performing model (Scaled BT + ExPO) is slightly better in terms of overall RewardBench (94.1 vs. 93.8). However, looking more closely at the individual categories, our best model does better in all categories (Chat, Safety and Reasoning) except Chat-Hard. On Chat-Hard, it trails substantially behind Skywork-Reward-Gemma-2-27B (85.7 vs. 91.4). To understand possible reasons for this, we looked more closely at the constituent sources of data for the Chat-Hard

category. We discovered that the Chat-Hard category is composed of data from two sources - LLMBar (Zeng et al., 2024) and MT-Bench (Zheng et al., 2023). LLMBar contains five different subsets that are used in Chat-Hard, three of which are based on human annotations as ground-truth labels, while the other two (Adversarial-GPTInst and Adversarial-GPTOut) use GPT-4 annotations as ground-truth labels. Similarly, MT-Bench also involves using GPT-4 judgements as the ground-truth.

| Source | Chat-Hard | Human as Ground Truth | | | GPT-4 as Ground Truth | | |
| | | LLMBar-Adversarial | | LLMBar | LLMBar-Adversarial | | MT-Bench |
| *Subset* | | *manual* | *neighbor* | *natural* | *GPTInst* | *GPTOut* | *hard* |
|---|---|---|---|---|---|---|---|
| Scaled BT + ExPO | 85.7 | 76.1 | 88.8 | 95.0 | 87.0 | 72.3 | 75.7 |
| Skywork-Reward-Gemma-2-27B | 91.4 | 78.3 | 89.6 | 96.0 | 97.8 | 91.5 | 86.5 |

Table 2: Performance of Models on RewardBench Chat-Hard Category. Higher is better for each subset. Scaled BT + ExPO uses Llama-3.1-70B-Instruct as a base model and initialized using a Helpfulness-Only SteerLM Regression training while numbers for Skywork-Reward-Gemma-2-27B are taken from RewardBench leaderboard (AllenAI, 2024)

As shown in Table 2, we found that our best model performs similarly (within 2.2% difference) to Skywork-Reward-Gemma-2-27B on subsets that use human annotations as ground-truth while doing much worse (10.8 - 19.2 %) on subsets that use GPT-4 judgements as ground-truth. One possible explanation is that the constituent dataset for the training data of Skywork-Reward-Gemma-2-27B contains GPT-4 annotated data and hence can better model GPT-4 judgements. We find evidence of this possibility from Skywork-Reward-Gemma-2-27B's training data description (Liu & Zeng, 2024) as it claims that the training data blend contains the Offset Bias dataset (Park et al., 2024), which has data as judged by GPT-4. Interestingly, the creators of LLMBar (Zeng et al., 2024) and MT-Bench (Zheng et al., 2023) have also noted the tendency of these tasks to overly favor models trained on GPT-4 data. Overall, this implies that our best model performs on par with the top external baseline model on the Chat-Hard category when biases associated with training on GPT-4 judgements are accounted for.

| | Aspects | | | Application Settings | | |
| | Accuracy | Interpretability | Inference Speed | Data-Filtering | RLHF | Human-in-Loop |
|---|---|---|---|---|---|---|
| SteerLM Regression | ✓ | ✓ | ✓✓ | ✓✓ | ✓ | ✓ |
| Bradley-Terry | ✓✓ | ✗ | ✓✓ | ✓ | ✓✓ | ✗ |
| Pairwise Justifier | ✗ | ✓✓ | ✗ | ✗ | ✗ | ✓✓ |

Table 3: Comparisons of different types of Reward Models in terms of aspects that they are strong on and application settings they are suitable for.

**When to use each type of reward models?**    While much of the discussion above compares reward model types in terms of their RewardBench overall accuracy, real-world use cases for Reward Models have many other considerations. We analyze a few of these aspects and how such considerations influence suitable application areas for each type of Reward Model, as illustrated in Table 3.

In terms of accuracy, Bradley-Terry models are the highest (when initialized on Helpfulness Only SteerLM Regression models), closely followed by SteerLM Regression models and lastly Pairwise Justifier models. Pairwise Justifier models are the most interpretable as they can generate an explanation of why they prefer a response over another. SteerLM Regression models are also somewhat interpretable as they can be used to predict the attribute values for various dimensions of a response (e.g. high in verbosity but low in complexity). Another factor which makes SteerLM Regression models interpretable is that the predicted scores are well-calibrated. The scores are typically within the range provided in the training data - in this case between 0 to 4, with each score having a real world meaning - *e.g.* 4 means perfectly helpful, 3 means mostly helpful and 0 means not helpful. However, Bradley-Terry Models are not calibrated which means there's no set interval for predictions. In the case of the Scaled BT + ExPO model, the range for rewards on RewardBench responses was [-33.50, 2.654]. BT Reward Scores also cannot be seen in isolation but must be interpreted in the context of the scores of other responses to the same prompt (see Appendix G). Finally, SteerLM Regression and Bradley-Terry are both extremely fast for inference - requiring the equivalent of 1 generated token in terms of compute while Pairwise Justifier models can require hundreds of tokens.

Because of these characteristics, different types of reward models are likely useful for different applications. SteerLM Regression is most suited for filtering SFT data (Nvidia et al., 2024) because data can be filtered both based on an absolute score threshold or relative score (*i.e.* highest score among 10 responses to the same prompt). On the other hand, Bradley-Terry models can only be used with relative score filtering. Bradley-Terry models are most suited for RLHF given that they are the most accurate with SteerLM Regression a close second. Finally, Pairwise Justifier models can be most suitable in supporting Human-in-the-Loop evaluation, where the explanation provided by the models can assist humans making a final decision. SteerLM Regression models can also do this by providing attribute values to dimensions of the response (e.g. complexity, correctness), but its format is less flexible for supporting human interpretation.

## 5 ALIGNED MODELS

To demonstrate the usefulness of our best reward model and HelpSteer2-Preference dataset in aligning language models to be helpful, we use them in three popular alignment algorithms.

### 5.1 EVALUATION

Following Wang et al. (2024d); Dong et al. (2024); Meng et al. (2024), we use three metrics to measure aligned models' ability to be helpful in responding to user prompts: GPT-4-Turbo MT Bench (Zheng et al., 2023), AlpacaEval 2.0 Length Controlled (Dubois et al., 2024) and Arena Hard (Li et al., 2024). We report the mean number of characters in MT Bench responses to give a sense of response length. MT Bench is also referenced as a validation metric for checkpoint selection. Further details are in Appendix H.

### 5.2 TRAINING

We use the Llama-3.1-70B-Instruct model (Dubey et al., 2024) to initialize the policy model for all experiments and Scaled BT + ExPO (94.1% RewardBench) as the reward model for PPO and REINFORCE. Training hyperparameters and the associated search range are in Appendix E.

**Direct Preference Optimization (DPO)**    Following Section 3.2, we transform the three variants of Bradley-Terry into corresponding DPO objectives: Regular DPO (Rafailov et al., 2023) corresponds to Eq. 1, Margin DPO (Touvron et al., 2023) to Eq. 2, and Scaled DPO to Eq. 3. We train models using the HelpSteer2-Preference dataset.

**Proximal Policy Optimization (PPO)**    Following the RLHF recipe prescribed by Ouyang et al. (2022), we align the policy model via PPO (Schulman et al., 2017). We initialize the value model with the reward model. We found it useful to run 2 rounds on PPO, where we pick the best checkpoint in round 1 to initialize the policy/reference models for Round 2. The value model is always reinitialized with the reward model at each round. Our training uses the HelpSteer2-Preference prompts.

**REINFORCE**    We align the policy model with REINFORCE (Williams, 1992). Following Ahmadian et al. (2024), we use the KL-regularized reward and employ the leave-one-out baseline, sampling four responses per training prompt (Kool et al., 2019). We train on HelpSteer2-Preference prompts.

### 5.3 RESULTS

As shown in Table 4, most attempted algorithms improve relative to Llama-3.1-70B-Instruct, demonstrating the strength of the HelpSteer2-Preference dataset and trained reward model.

**Offline vs. Online RLHF**    Across three DPO variants, there is consistent improvement over the base Llama-3.1-Instruct model in terms of GPT-4-Turbo MT-Bench as well as AlpacaEval 2.0 LC. We find that Scaled DPO performs best, underscoring the importance of adequately modelling the preference strength information we collected. However, no version of DPO can beat PPO or REINFORCE (across any of the three alignment metrics), highlighting the importance of using online training along with reward information. We report ablation studies using alternative reward model in Appendix L.

| Model Type | Model | Aligned Metrics | | | | Hyperparams | |
|---|---|---|---|---|---|---|---|
| | | MT Bench (GPT-4-Turbo) | Mean Response Length (Chars.) | AlpacaEval 2.0 LC (SE) | Arena Hard (95% CI) | LR | KL |
| **_Offline RLHF_** | Regular DPO | 8.66 | 1502.2 | 40.4 (1.66) | 52.8 (-2.7, 2.7) | 2e-7 | 0.01 |
| | Margin DPO | 8.58 | 1496.6 | 41.1 (1.67) | 52.6 (-2.7, 2.8) | 2e-7 | 0.001 |
| | Scaled DPO | 8.74 | 1514.8 | 41.0 (1.68) | 52.9 (-2.4, 3.1) | 2e-7 | 0.001 |
| **_Online RLHF_** | PPO | 8.74 | 1842.8 | 43.8 (1.76) | 58.6 (-2.9, 2.5) | 1e-6/1e-7 | 0.005/0.01 |
| | REINFORCE | **8.98** | 2199.8 | **57.6** (1.65) | **85.0** (-1.5, 1.5) | 5e-7 | 0.01 |
| **_External Baselines_** | Llama-3.1-70B-Instruct | 8.22 | 1728.6 | 38.1 (0.90) | 55.7 (-2.9, 2.7) | | |
| | Llama-3.1-405B-Instruct | 8.49 | 1664.7 | 39.3 (1.43) | 69.3 (-2.4, 2.2) | | |
| | Claude-3-5-Sonnet-20240620 | 8.81 | 1619.9 | 52.4 (1.47) | 79.2 (-1.9, 1.7) | | |
| | GPT-4o-2024-05-13 | 8.74 | 1752.2 | 57.5 (1.47) | 79.3 (-2.1, 2.0) | | |

Table 4: Performance of Aligned Models. Higher is better for each metric, except Length. All models are trained by us using Llama-3.1-70B-Instruct as a base model except External Baselines, the scores for which are taken from Arena Hard (LMSys, 2024) and AlpacaEval leaderboards (Tatsu-Lab, 2023)

**Variants of Online RLHF: PPO vs. REINFORCE**    Both PPO and REINFORCE can improve performance, with REINFORCE showing a marked advantage. Similar to Ahmadian et al. (2024), we find that REINFORCE is much better at maximizing the reward than PPO (See Appendix I for reward curves). We hypothesize that this is due to differences in baseline estimation in PPO and REINFORCE. In PPO, a learned critic function is used to estimate state values, which can introduce bias and instability (Kumar et al., 2020). In contrast, for REINFORCE we use a leave-one-out baseline estimator – an unbiased and stable Monte-Carlo estimator of the policy's value function (Sutton, 2018). This leads to more stable training, and allows REINFORCE to outperform PPO.

**Comparison to Frontier Models**    Our best model (trained with REINFORCE) achieves competitive performance with frontier models such as GPT-4o and Claude 3.5 Sonnet across the three popular alignment benchmarks (MT Bench, AlpacaEval 2.0 LC and Arena Hard). This highlights the strength of our Reward Model to guide online RLHF. REINFORCE significantly increases the response length compared to the base models, but its high length-controlled AlpacaEval 2.0 score suggests that the extra tokens are meaningful. However, it is important to bear in mind the limitations of these automated evaluations, as all of them use a LLM-as-a-judge (GPT-4-Turbo), which is generally perceived as weaker than the top-performing External Baselines like GPT-4o and Claude 3.5 Sonnet. As a result, the LLM-judge might not be able to adequately evaluate the quality of such model responses, in relation to responses generated by our aligned models. This means that such automated evaluation for alignment may be insufficient to inform on how our models perform in relation to such frontier models in diverse use cases. Given that our main purpose is to demonstrate how our trained reward models can be useful for RLHF, we leave more holistic evaluations to Appendix M.

**Case Study**    We adopt a prompt that many have recently been vibe-testing LLMs with - `How many r in strawberry?` Among models in Table 5, only REINFORCE can correctly answer it.

| Model | Response to *"How many r in strawberry?"* |
|---|---|
| Regular / Margin / Scaled DPO | There are 2 r's in the word "strawberry". |
| PPO | There are 2 R's in the word "strawberry". |
| **REINFORCE** | A sweet question! Let's count the "R"s in "strawberry": 1. S 2. T 3. R 4. A 5. W 6. B 7. E 8. R 9. R 10. Y There are **3 "R"s** in the word "strawberry". |
| Llama-3.1-70B-Instruct | There are 2 R's in "strawberry". |
| Llama-3.1-405B-Instruct | There are 2 Rs in the word "strawberry". |
| Claude-3-5-Sonnet-20240620 | There are two "r" letters in the word "strawberry": strawberry The first "r" appears after "st" and the second "r" is part of the "rry" at the end of the word. |
| GPT-4o-2024-05-13 | There are two "r" letters in the word "strawberry." |

Table 5: Model responses to "How many r in strawberry?" Newlines in responses not shown.

## ETHICS STATEMENT

The prompts and responses in the HelpSteer2-Preference dataset do not contain any unsafe content (e.g. harmful content, illegal activities, profanity, bias and stereotyping) or any content containing Personally Identifiable Information (e.g. name, address, SSN, email, phone numbers), which were excluded by the HelpSteer2 (Wang et al., 2024d) collection effort. Annotators who supported the construction of the dataset were contracted through Scale AI, which completed ethical review prior to the start of data collection. Scale AI engages the Anker Methodology, GISC Impact Sourcing Standard, and UN Sustainable Development Goals to provide a fair and competitive pay. The specific pay is calculated based on many factors, including the specific project, the specialized skillset and expertise required, regional costs of living and then transparently listed on Scale AI platform. Scale AI also provides multiple channels for questions and support, including 24/7 support teams, community discussion channels with specially trained moderators, and a "speak up" hotline where contractors can report concerns anonymously. Worker concerns can be submitted to and are reviewed by the Remotasks support team, and pay disputes are reviewed by support specialists trained in this area.

## REPRODUCIBILITY STATEMENT

**Data Pre-processing:** Sec. 2 and Appendix C

**Training Hyper-parameters:** Appendix E

**Training Compute:** Appendix F

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

## A    RELATED WORKS

**Eliciting Human Judgements using Preference Rankings vs. Ratings**    There has been a series of work from sociological statistics on comparing the relative merits of ranking and rating style questionnaires in understanding human judgements. Alwin & Krosnick (1985) found both ranking and rating can perform similarly in terms of modelling overall preference but capture different (but complementary) aspects of their preference. McCarty & Shrum (2000) similarly found that forced-choice ranking yielded more differentiated preference data compared to Likert-type ratings, with the additional advantage of fewer extreme scores. We build upon this literature to examine whether the same differences can be observed in collecting domain-general data for preference tuning of LLMs.

**Bradley-Terry style Preference Datasets**    HH-RLHF (Bai et al., 2022) was the first open-source general-domain, human-annotated preference dataset released by Anthropic, containing over 160,000 pairs of responses and annotations on which one response is preferred amongst the pair. However, there has been many concerns relating to the quality of this data (Wang et al., 2024c; Shen et al., 2024) There are many more domain-specific datasets for individual tasks such as long-form question answering: OpenAI WebGPT Nakano et al. (2021)), summarization: OpenAI Summarize (Stiennon et al., 2022), online forum responses: Stanford Human Preferences Dataset (Ethayarajh et al., 2022), but they are less useful for modelling general-domain preferences.

**Regression-style Preference Datasets**    OpenAssistant (Köpf et al., 2023) is a prominent crowdsourced, open-source general domain human-annotated dataset with >10,000 conversation trees released by the Open Assistant organization. Open Assistant contains Likert-5 annotations for various attributes including helpfulness, creativity, and humor. Ultrafeedback (Cui et al., 2023) is a GPT-4 annotated dataset containing 64,000 prompts each with 4 responses. For each response,

there are 4 attributes annotated on a Likert-10 scale: helpfulness, honesty, instruction-following, and truthfulness. With each rating, there is also a short (2-3 sentences) rationale that explains the rating. Ultrafeedback has previously been converted into a preference dataset (Tunstall et al., 2023) based on the mean of the 4 attribute scores - with the highest scoring response as the chosen and one of the other three responses selected at random as the rejected. HelpSteer (Wang et al., 2024e) is a human-annotated general domain dataset with 10,000 prompts, each with 4 responses. Each response is labelled by 1 annotator on five Likert-5 attributes - helpfulness, correctness, coherence, complexity, and verbosity. However, Shen et al. (2024); Wang et al. (2024d) found the above datasets to substantially noisy, which means that reward models are unable to different between responses are not drastically different in quality. HelpSteer2 (Wang et al., 2024d) is a follow-up to HelpSteer with 10,000 real-world prompts, each with 2 responses. Each response is annotated on the same five Likert-5 attributes. However, every response is annotated by 3-5 annotators with the final ratings obtained by removing outlier annotations and then taking the average of the remaining annotations.

# B  PREFERENCE RANKING GUIDELINES

## B.1  RANKING PRIORITIZATION

Ratings should be made by prioritizing the response characteristics below in the order they are stated. Trade-offs should favor higher priority characteristics.

1. Instruction Following

    - Responses should follow all requests in the prompt, following the instructions in the prompt is the primary criteria for ranking responses.
    - Many prompts will have a clear instruction that has additional or implicit requirements, e.g. "plan an itinerary for a 5-day trip to Portugal that focuses on surfing" should be evaluated on whether an itinerary is produced, whether it is 5 days long and in Portugal, and also whether it includes a lot of surfing - All instructions should be considered when ranking responses, with the core instructions being weighed more heavily.
    - If specific formatting (table, json) is asked for in a prompt that should be considered as the high priority "instruction following" and not general "formatting" which is lower priority.

2. Correctness

    - Answers which are factually correct should be rated higher than answers that are incorrect.
    - Annotators should search the internet when they are unsure about correctness.
    - Misleading or incomplete answers are considered less correct than complete answers.
    - If prompt (question) contains false premise, the response that pushes back on it should be preferred.
    - When a question cannot be answered definitively, the response which expresses uncertainty should be preferred.

3. Formatting

    - When no specific formatting is requested responses with better formatting are preferred.
    - Vendor tool should render markdown for easier assessment of formatting (markdown tables, lists, shell scripts, etc. should be rendered properly).
    - Formatting encompasses anything that is appropriate for the response, e.g. an itinerary being split by day, including a markdown table when it makes sense but was not asked for, any appropriate use of markdown, responding with a shell script, etc.

4. Clarity

    - Answers that are easier to understand, provide more clarity (via explanation or introduction), or are generally better written are preferred.
    - Unnecessary verbosity, or providing extra information that is irrelevant should be penalized.

## B.2 RANKING STRENGTH

We would like to collect three levels of preference, described below:

1. Response 1 is slightly better than Response 2 (or vice-versa)

   - To be used when the responses are similarly appropriate and the difference is minor or a matter of personal preference.
   - There should be no difference in Instruction Following or Correctness if this is selected.
   - Minor differences in clarity and formatting warrant this response.
   - When the annotator considers the options to be tied in all other aspects, they should prefer the shorter answer (in unlikely circumstances of same length - use their own judgment)

2. Response 1 is better than Response 2 (or vice-versa)

   - To be used when one response is clearly better than the other, such as:
   - If one response is clearly better than the other, but the worse response is still appropriate (it is not incorrect or off topic).
   - If the worse response fails to follow some instructions or is correct but not well phrased/structured, and the better response also fails some instructions at the same level, but has no other flaws overall.
   - If the better response fails to follow some but not all instructions or is misleading but the worse response does not follow instructions at all or is wrong, this should be selected.
   - If both answers follow instructions and are correct, but one is clearer and/or better formatted, this should be selected.

3. Response 1 is much better than Response 2 (or vice-versa)

   - To be used when there is a significant difference between the two responses.
   - If one answer is correct and the other isn't, this should be selected.
   - If one answer follows all instructions and the other does not follow instructions, this should be selected.
   - If there are major differences in readability and formatting, this should be selected.

4. Neither response is valid

   - If neither response follows the instructions or provides a correct answer, this option should be chosen.
   - This is meant to be used only for egregious issues. If both answers could be improved, but they follow what the prompt asked with no critical error then a preference should be selected instead.

## B.3 RANKING EXAMPLES

*Note that the full guidelines contain more examples, but we show here only four illustrative ones, one for each preference strength.*

### B.3.1 RESPONSE 1 IS SLIGHTLY BETTER THAN RESPONSE 2

**Prompt**
Describe the process of photosynthesis. Additionally, explain how variations in light intensity can affect this process and how plants have adapted to such changes.

**Response 1**
Photosynthesis is the process by which green plants and some other organisms convert light energy, typically from the sun, into chemical energy in the form of glucose (sugar). It can be broken down into two stages:
1. Light-dependent reactions: These occur in the thylakoid membranes of chloroplasts. Light energy is absorbed by chlorophyll and other pigments, splitting water molecules and producing oxygen, electrons, and ATP (adenosine triphosphate).

2. Light-independent reactions (Calvin cycle): These occur in the stroma of the chloroplast. Carbon dioxide is taken in from the atmosphere, and with the help of ATP and electrons produced in the light-dependent reactions, it is converted into glucose and other organic molecules.

Variations in light intensity can affect photosynthesis in the following ways:
- Low Light Intensity: Photosynthesis decreases as there is insufficient light for the light-dependent reactions. This leads to a lower production of ATP and NADPH, slowing down the Calvin cycle and reducing glucose production.
- High Light Intensity: Excessive light can damage chlorophyll and other pigments, leading to reduced photosynthesis. To protect themselves, plants have adapted by mechanisms like photoprotection and non-photochemical quenching to dissipate excess light energy as heat.

Plants have evolved several adaptations to cope with changes in light intensity, including adjusting the size of their chloroplasts, altering the angle of their leaves, and using pigments that can capture light at different wavelengths. Additionally, they can regulate their stomata to control the entry of carbon dioxide and reduce water loss during times of intense sunlight. These adaptations help optimize photosynthesis under varying light conditions.

**Response 2**
Photosynthesis occurs in chloroplasts of plant cells. It uses sunlight, carbon dioxide, and water to produce glucose and oxygen. Light energy is absorbed by chlorophyll, splitting water molecules and releasing oxygen. The energy from light is used to convert carbon dioxide into glucose.
Higher light intensity increases the rate of photosynthesis, as more light energy drives the light-dependent reactions. However, other factors like CO2 and temperature can limit the rate. Too much light can damage the photosynthetic machinery.
Plants adapt to varying light through changes in leaf angle, altering chlorophyll production, opening/closing stomata, and photorelocation of chloroplasts. These adaptations maximize photosynthesis in both low light and intense light conditions by balancing light capture with CO2 intake and production capacity.

**Preference Justification**
@Response 1 is slightly better than @Response 2. While both responses correctly answers the prompt, the style and formatting of the better response is preferred. It also gives additional facts about photosynthesis.

### B.3.2    RESPONSE 1 IS BETTER THAN RESPONSE 2

**Prompt**
Discuss the benefits of renewable energy sources.

**Response 1**
Renewable energy sources such as solar, and wind offer numerous benefits to mitigate climate change. Unlike exhaustible fossil fuels, renewables rely on naturally replenished resources. Transitioning to renewables will significantly benefit public health and the environment by reducing hazardous air and water pollution produced from extracting and burning fossil fuels.

**Response 2**
Renewable energy sources, such as solar, wind, and hydro power, offer numerous benefits. They produce clean energy, reducing greenhouse gas emissions and our reliance on fossil fuels. Additionally, renewable energy is sustainable, helping to protect the environment and conserve resources for future generations.

**Preference Justification**
@Response 1 is better than @Response 2. Both responses approach the prompt from slightly different angles. One could consider @Response 1 the better response given its additional information about transition to renewable sources and how it could impact the environment.

### B.3.3    RESPONSE 1 IS MUCH BETTER THAN RESPONSE 2

**Prompt**
Make an itemized list of 4 elements with actions that can increase the emission of oxygen. Item markers must be lower case letters starting from D.

**Response 1**

d. Planting trees: Trees release oxygen through the process of photosynthesis, making them the most effective way to increase oxygen levels.

e. Growing algae: Algae in ponds or water bodies can play a significant role in oxygen production. Cultivating algae can increase their growth and subsequent oxygen release.

f. Creating green spaces: Designating and maintaining green spaces in urban areas, such as parks and gardens, help support plantations and increase oxygen levels.

g. Promoting indoor plants: Placing indoor plants in your living or working space can enhance oxygen production and improve air quality, benefiting the overall oxygen levels indoors.

**Response 2**

D. Photosynthesis: Photosynthesis is the process by which plants, algae, and some bacteria convert light energy from the sun into chemical energy in the form of organic compounds, such as glucose. This process releases oxygen as a byproduct, which is then released into the atmosphere.

E. Respiration: Respiration is the process by which cells in living organisms convert glucose into energy. During this process, oxygen is consumed and carbon dioxide is produced. Increasing the rate of respiration in living organisms can increase the amount of oxygen released into the atmosphere.

F. Exercise: Exercise is a physical activity that can increase the rate of respiration in the body. As the body works out, it burns more glucose and consumes more oxygen, leading to an increase in oxygen emissions.

G. Forestation: Planting more trees and forests can increase the amount of photosynthesis occurring in an area, which can lead to an increase in oxygen emissions. Trees and other vegetation absorb carbon dioxide from the atmosphere and release oxygen as a byproduct of photosynthesis.

**Preference Justification**

@Response 1 is much better than @Response 2. The prompt specified that the item markers should be lower case. @Response 2 starts from the letter "D", but it is not lower case. In addition, @Response 2 has incorrect facts about respiration and is not coherent/fluent regarding the prompt.

### B.3.4    NEITHER RESPONSE IS VALID

**Prompt**

When it is 1pm in San Francisco, what time is it in Singapore?

**Response 1**

It is 8pm in Singapore when it is 1pm in San Francisco.

**Response 2**

It is 10pm in Singapore when it is 1pm in San Francisco.

**Ranking explanation**

Neither option is factually correct. The correct answer depends on whether Daylight Saving Time is in effect. When it is 1 pm in San Francisco during Standard Time, it is 5 am the next day in Singapore. During Daylight Saving Time in San Francisco, when it's 1 pm in San Francisco, it is 4 am the next day in Singapore.

## C    JUSTIFICATION PRE-PROCESSING

Our annotation guidelines provided minimal guidance on how the preference justification should be written within the free-text box. One such guidance was that annotators should consistently refer to the rated responses as "@Response 1" and "@Response 2". This meant preference justifications do not always conform to a fixed format. To better understand these justifications, we split them into sentences using NLTK sentence-tokenizer (NLTK, 2024) as well as newlines, to facilitate subsequent post-processing.

Many preference justifications contain a preference statement, in the form "@Response 1 (or 2) is better ... ". Most commonly (57.6% of samples), these preference statements are only contained in the first sentence. This is followed by 27.5% of samples containing it in both the first sentence and the last sentence. A minority (7.3%) contains it in the last sentence while a similar proportion (7.6%) does not contain such a preference statement at all. Among those without a preference statement, a qualitative analysis revealed that some state the strengths and weakness of each response without comparison,

some claim that both responses are equal (despite our schema not supporting equivalence) while others make aspect-specific comparisons without stating if it makes either response better.

We believe that it is important to have all justifications in a similar format because use cases such as LLM-as-a-judge require justifications to be standardized for automatic parsing. Therefore, we split the justification into a preference statement and a preference elaboration. This enables users to freely decide if they prefer to generate the preference statement first or the preference elaboration first. For justifications containing a preference statement both at the front and the back, we remove the trailing preference statement to avoid duplication. In addition, preference statements that are extracted from the end of justifications are frequently prefixed with terms like "Therefore, ", "Overall, " and "For these reasons, ". To ensure uniformity, we remove all prefixes before the first "@Response 1/2 ... better' is encountered.

Requiring all preference justifications to have a preference statement makes justifications without one difficult to use directly. After considering several approaches (e.g. prompting an LLM to restructure justifications to follow a format or manually adding in the preference statement using the associated preference score), we decided that the best solution would be to exclude the preference justifications without a preference statement altogether. This is because the potential benefit of having an additional 7.6% of data was not worth the risk of LLM hallucinations or potentially contradictory/repetitive information from the rule-based heuristic.

Finally, we prepare a subset of these justifications into the Overall Preference Justification, by randomly selecting one justification per task, whose annotator has given the same individual preference as the overall preference. The Overall Preference Justification set, which comprises 6, 618 samples with 6, 287 in the train set and 331 in the validation set while the All Justifications dataset has 24, 698 justifications (23, 487 train and 1, 211 val).

| Attribute | All Justifications | | Overall Preference Justifications | |
|---|---|---|---|---|
| | Statement (Std.) | Full (Std.) | Statement (Std.) | Full (Std.) |
| No. of justifications | 24698 (incl. tied preference) | | 6618 | |
| Justifications per task | 2.71 (0.51) | | 1 (0) | |
| No. of sentences | 1 (0) | 3.95 (2.11) | 1 (0) | 4.03 (2.10) |
| No. of characters | 105.9 (68.3) | 483.3 (266.3) | 130.8 (72.7) | 494.5 (258.5) |
| Prefers Response 1 (%) | 47.1 | | 46.4 | |
| Prefers Response 2 (%) | 52.9 | | 53.6 | |
| Contains 'because' (%) | 46.8 | 53.0 | 47.4 | 53.8 |
| Mentions Response 1 Only (%) | 7.9 | 0.9 | 7.5 | 0.7 |
| Mentions Response 2 Only (%) | 8.9 | 1.3 | 8.7 | 1.1 |
| Mentions both Responses (%) | 83.2 | 97.8 | 83.7 | 98.2 |

Table 6: Descriptive statistics for preference justifications. Preference justification consists of both a preference statement and a preference elaboration, and both together is termed *Full*. Overall Preference Justifications is a subset of all justifications in which only one representative justification is used for each task.

## D  PREFERENCE JUSTIFICATION ANALYSIS

In Table 6, we analyze the descriptive statistics for preference justifications. On average, preference justifications contain 4 sentences or 500 characters (approximately 100 words). There is a slight preference for Response 2 over Response 1 (6-7%). Approximately half of all preference statements contain the word "because" while other words implying rationales such as 'due to' and 'owing to' are minimal ($<1\%$). The other half of the preference statement communicates a preference without explanation. The vast majority of responses (83%) mention both responses in the preference with nearly all responses mentioning both responses in the full justification. This suggests that most annotators used a comparative approach between two responses, indicating that they follow our guidelines well.

To better understand the content contained in preference justifications, we conducted a word-level analysis of Overall Preference Justifications (including both preference statements and preference elaborations). We first lowercase the justifications and replace all non-alphanumerical symbols with

| Attribute | List of attribute-relevant keywords / factors (% of occurrence) | % of Justifications | |
| --- | --- | --- | --- |
| | | w. keywords | LLM-classified |
| **_HelpSteer Attributes_** | | | |
| Helpfulness | **All:** help, helpful, helpfulness, instruction, unhelpful, useful | 11.0 | 77.5 |
| Correctness | **Positive:** accurate, accurately, complete, correct, factual, informative
**Negative:** error, false, inaccurate, incomplete, incorrect, incorrectly, misses, missing, wrong
**Neutral:** completeness, correctness, fact, information, understand, understanding | 12.9 | 61.4 |
| Coherence | **Positive:** clear, clearer, direct, directly, relevant
**Negative:** confusing, irrelevant, redundant, repeats, repetitive, unclear, unnecessary, vague
**Neutral:** clarity, coherence, structure
**Format:** bulleted, format, formatted, list, listed, numbered, outline | 3.5 | 30.3 |
| Complexity | **All:** basic, depth, difficult, easier, easy, simple, simply | 0.6 | 8.0 |
| Verbosity | **Short:** brief, concise, short, shorter, succinct,
**Long:** comprehensive, detailed, long, longer, thorough, verbose,
**Neutral:** detail, details, length, verbosity | 4.0 | 23.6 |
| **_Other Factors_** _(Sharma et al., 2023)_ | | | |
| Stylistic | **Nice:** friendly (1.4), polite (0.6), empathetic (0.5), optimistic (0.1)
**Joyful:** engaging (2.7), entertaining (1.0), funny (0.5)
**Charismatic:** persuasive (1.0), authoritative (0.3), motivating (0.2) | - | 5.9 |
| Sycophantic | **All:** match_human_style (1.0), agree_human_explicit (0.3), agree_human_implicit (0.03) | - | 1.4 |
| HelpSteer-Adjacent | **Helpfulness:** relevant (36.7), well_written (38.0)
**Correctness:** informative (63.7), truthful (16.0), better_supported (13.3), rigorous (22.3)
**Coherence:** structured (13.5), grammatically_sound (1.5), logically_sound (9.8)
**Complexity:** higher_reading_age (0.5)
**Verbosity:** concise (19.7) | - | 90.1 |

Table 7: Analysis of keywords and factors mentioned in preference justifications.

empty strings, before splitting justifications into bags of whitespace-separated words. We then count the number of justifications in which each word appears. From the top 500 most frequent words, we manually identify those that relate to each HelpSteer2 attribute.

Based on Table 7, we found that annotators used each of the five HelpSteer2 attributes to guide their justification writing. Most influential was Correctness-related words (12.9%) followed by Helpfulness-related words (11.0 %). We suspect that the Helpfulness is under-represented because very few keywords (6) are closely associated with helpfulness while many more (21) are associated with correctness. Surprisingly, verbosity is the next most influential to annotators (4.0 %), and more so compared to coherence (3.5 %). We hypothesize that this might be due to responses from HelpSteer2 being mostly Coherent (mean of 3.64 out of 4) meaning that both responses are likely to be perfectly coherent. Finally, complexity-related words only appear in 0.6% of justifications, suggesting that they barely affect preference judgments. Overall, the proportion of justifications that can be explained by these attributes however, is low at ($<$32.0%). This is likely due to our word-level analysis only involving the top 500 most frequent keywords, which might not be sufficient to capture all aspect-relevant features. For instance, an annotator might communicate that a response is more helpful than another, using the terms helpful, useful, or instruction (following).

To complement the word-level analysis, which is interpretable but with low recall, we also prompted an LLM, specifically Nemotron-4-340B-Instruct (Nvidia et al., 2024) to classify whether the preference justification discusses each attribute (prompts in Table 8). LLM-based analysis shows a much higher proportion of justifications mentioning each attribute. Nonetheless, the relative proportion between attributes remains similar - with the primary factors influencing preference being helpfulness and correctness; followed by coherence and verbosity; and complexity trails behind as the least important attribute.

In addition, we wanted to understand if factors outside of HelpSteer attributes influenced annotator preferences. We used a list of 24 factors identified by Sharma et al. (2023) including politeness, empathy, and persuasiveness, and prompt the LLM to classify in a similar manner. After classifying each factor separately, we organized them into three main categories - Stylistic, Sycophantic (i.e. similar to the prompt), and HelpSteer-Adjacent. Unsurprisingly, HelpSteer-Adjacent factors are most commonly mentioned in these justifications (90.1%). Comparatively, stylistic factors and sycophantic factors that were found to substantially influence human and LLM judgments in previous work (Sharma et al., 2023; Chiang et al., 2024) are rarely discussed by our annotators (5.9% and 1.4%). This suggests the effectiveness of our guidance to annotators on focusing on the 'substance' of responses.

| Attribute | Question |
|---|---|
| **HelpSteer Attributes** | |
| helpfulness | the helpfulness or understanding of the response(s)? |
| correctness | the correctness or completeness of the response(s)? |
| coherence | the coherence or clarity of the response(s)? |
| complexity | the complexity or simplicity of the response(s)? |
| verbosity | the verbosity or succinctness of the response(s)? |
| **Other Factors** *(Sharma et al., 2023)* | |
| *Stylistic* | |
| friendly | how friendly the response(s) were? |
| polite | how polite the response(s) were? |
| empathetic | how empathetic the response(s) were? |
| optimistic | how optimistic the response(s) were? |
| engaging | how engaging the response(s) were? |
| entertaining | how entertaining the response(s) were? |
| funny | how funny the response(s) were? |
| persuasive | how persuasive or compelling the response(s) were? |
| authoritative | the authoritativeness or assertiveness of the response(s)? |
| motivating | how motivating the response(s) were? |
| Sycophantic | |
| match_human_style | how well the response(s) match the prompt's writing style? |
| agree_human_explicit | whether either the response(s) agree with the preferences, biases, and beliefs explicitly stated by the prompt? |
| agree_human_implicit | whether either the response(s) agree with the preferences, biases, and beliefs implied by the prompt? |
| HelpSteer-Adjacent | |
| relevant | how relevant the response(s) were to the prompt? |
| well_written | how well the response(s) were written? |
| informative | how informative the response(s) were? |
| truthful | how truthful the response(s) were? |
| better_supported | how well-supported the response(s) were? |
| rigorous | how rigorous the response(s) were? |
| structured | how well-structured the response(s) were? |
| grammatically_sound | the grammatical soundness of the response(s)? |
| logically_sound | how logically sound the response(s) were? |
| higher_reading_age | the reading age that the response(s) were written for? |
| concise | how concised or focussed the response(s) were? |

Table 8: Questions used for analyzing if preference justification discusses each attribute or factor. Each question was used with the template '{justification} Does the above comparison between @Response 1 and @Response 2 discuss {question} Answer only with yes or no.'

## E  TRAINING HYPER-PARAMETERS

**Reward Modelling**  For SteerLM Regression models, we used 2 epochs of HelpSteer2 data, following Wang et al. (2024d). For Bradley-Terry models, we used 1 epoch of HelpSteer2-Preference data, as more than 1 epoch resulting in overfitting - similar to as observed by Zhu et al. (2024) and drastically higher validation loss and poorer RewardBench performance. For Pairwise Justification models, we train for 1 epoch for each setup - we also tried 2 epochs for initial experiments which showed minimal changes from only training on 1 epoch. For each experiment, we used global batch size of 128 using an AdamW optimizer with 10 warm-up steps and performed search over constant learning rates of $\{1, 2, 3\}e - 6$. We report the optimal learning rate for each experiment in Table 1. We save checkpoints every 10 steps and evaluate three checkpoints with the lowest validation loss as

well as the last checkpoint. Among them, we report the score with the highest RewardBench overall score. For SteerLM Regression Model with all five 5 HelpSteer attributes, we do grid search over all five attributes (between -1 and 1 at intervals of 0.1) over RewardBench.

**Direct Preference Optimization**  We trained DPO models for 2 epochs using the AdamW optimizer, after initially experimenting with 1 and 3 epochs but finding 2 epochs to be optimal. We use a global batch size of 64 response-pairs, weight decay of 0.01, with checkpoints saved every 10 steps, and 10 warmup steps followed by a constant learning rate. For hyper-parameter tuning, we performed a grid search over learning rates of $\{1, 2, 3\}e - 7$ and KL penalties of $\{0.01, 0.001\}$. We report the optimal learning rate, KL penalty for each experiment in Table 4.

**Proximal Policy Optimization**  We trained PPO models for a total of 2 rounds using the AdamW optimizer weight decay of 0.1, constant learning rate schedule and 30 steps of value model warmup. Due to the instability of PPO training also observed by (Zhu et al., 2023), we save checkpoints every 2 steps with our final checkpoints being at step 26 and 40 for rounds 1 and 2 respectively. For hyperparameter tuning we performed a grid search over learning rates {5e-6, 1e-6, 5e-7, 1e-7, 1e-8}, KL penalties of {0.1, 0.01, 0.001, 0.005, 0.0001} and training global batch size of {64, 128, 256}. Compared to other algorithms, we performed more hyperparameter search because we found PPO to be more sensitive to hyperparameters. In our experiments we set the training global batch size to be the same as the rollout global batch size and found 128, 256 the best for rounds 1 and 2 respectively. We report the optimal learning rate, KL penalty for each experiment in Table 4.

**REINFORCE**  Similar to PPO, we trained REINFORCE models using the AdamW optimizer with rollout batch size of 64, training batch size of 64, weight decay of 0.1, and a constant learning rate schedule with ten warmup steps. We sample four samples per prompt, and use the leave-one-out baseline. We save checkpoints every 5 steps, and the best checkpoint is found at step 95. for hyperparameter tuning we performed a grid search over learning rates {1e-7, 3e-7, 5e-7, 1e-6} and KL penalties of {0.01, 0.1}. The optimal learning rate and KL penalty are reported in Table 4.

## F  COMPUTE REQUIREMENTS

| Model | Compute (H100-eqv. node-hours) |
|---|---|
| Reward Models | |
| SteerLM Regression | 24 |
| Bradley-Terry | 8 |
| Pairwise Justifier | 32 |
| Aligned Models | |
| DPO | 16 |
| PPO | 50 |
| REINFORCE | 64 |

Table 9: Compute required for training models, measured in H100-eqv. node-hours. Experiments are run on nodes of 8 A100/H100-80GB SXM GPUs on internal clusters. For ease of comparison, every three A100 node-hours is converted to one H100 node-hour.

## G  REWARDBENCH RESPONSE DISTRIBUTION

Fig. 2 shows the distribution of reward scores on Rewardbench responses as judged by our best Reward Model.

## H  ALIGNED MODEL EVALUATION DETAILS

**MT Bench**  We follow (Meng et al., 2024; Tenyx, 2024; Wang et al., 2024d) to use MT Bench (Zheng et al., 2023), with GPT-4-Turbo (specifically GPT-4-0125-Preview) as the judge. MT Bench consists

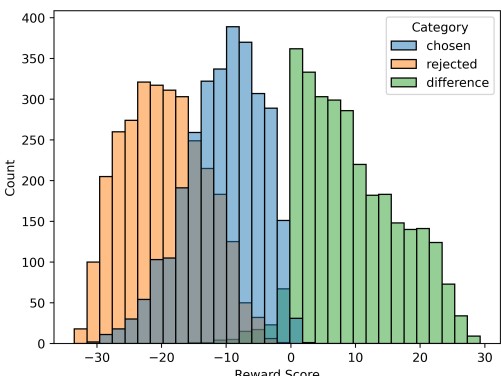

Figure 2: Distribution of Reward Scores for RewardBench responses by our best reward model (Scaled BT + ExPO, initialized on Helpfulness-Only SteerLM Regression model). Difference refers to the difference between the reward scores of chosen and rejected responses to the same prompt. When a response has reward of -15, it is equally likely to be chosen or rejected. This meaning the score of the response cannot be seen in isolation to determine if a response is good (or bad).

of 80 multi-turn questions, each consisting of an initial question and a follow-up question, for a total of 160 prompts. Questions are collated from 8 categories: Writing, Roleplay, Extraction, Reasoning, Math, Coding, STEM and Humanities/Social Science. We first greedily generate responses with up to 1024 tokens (default value for MT Bench). The responses to these prompts are evaluated by GPT-4-0125-Preview to give a score between 1 and 10, and we report the mean across all prompts. Prompts in Coding, Math and Reasoning categories are evaluated with a reference correct answer, which were generated by the judge model and then manually verified. Higher MT Bench score indicates better instruction following ability.

**AlpacaEval 2.0 Length Controlled** We follow Dong et al. (2024); Meng et al. (2024); Wang et al. (2024d) to use AlpacaEval 2.0 Length Controlled (Dubois et al., 2024). AlpacaEval 2.0 contains 805 first-turn instructions (relating to singular-requirement, straightforward tasks such as recommendations, question answering, and open-ended writing). An answer to each prompt is greedily generated by the evaluated model as well as a baseline model (GPT-4-1106-turbo), which are then sent to GPT-4-1106-turbo evaluator that outputs the probability of preferring the generations of the evaluated model. Finally, the authors introduced a length correction factor to mitigate the bias for the evaluator towards preferring longer generations.

**Arena Hard** We follow Dong et al. (2024); Meng et al. (2024); Wang et al. (2024d) to use Arena Hard (Li et al., 2024). Arena Hard contains 500 first-turn instructions obtained from challenging user queries on Chat Arena (Chiang et al., 2024). Prompts are classified using an LLM (Llama-3-70B-Instruct) to determine if they are complex, specific, real-world-application-related or require domain knowledge, problem-solving, creativity, technical accuracy. Prompts that meet at least 6 out of 7 criteria are labelled as challenging. Therefore, a huge proportion of prompts (>50%) are related to solving coding problems. Model responses are then compared with responses from GPT-4-0314 using GPT-4-1106-preview judge to calculate a win-rate over GPT-4-0314.

## I   REWARD CURVES

We plot the reward obtained by PPO and REINFORCE in Fig. 3.

## J   FURTHER ANALYSIS FOR SCALED BRADLEY TERRY

To give a clearer analysis for Scaled BT, we demonstrate the losses for various BT variants for representative scenarios in Table 10. Compared to Regular BT, Scaled BT increases the loss proportionally when the human-annotated margin is larger. Relative to Margin BT, Scaled BT provides a higher loss

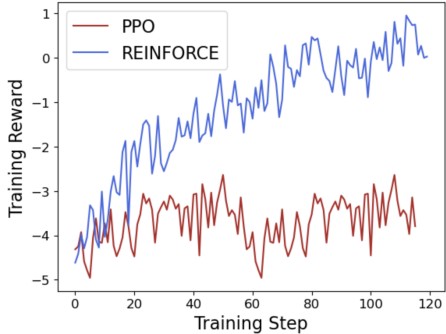

Figure 3: Reward curves of PPO and REINFORCE. Please note that REINFORCE samples four times as many responses as PPO per training step. For PPO, the reward curve is produced by concatenating the reward curves for Round 1 (ending at step 57) and Round 2. We select step 26 for PPO Round 1, step 40 for PPO Round 2 (whose reward is shown at step 97 above) while for REINFORCE, we select step 95.

from incorrect predictions while keeping losses low on correct predictions. Specifically, Scaled BT keeps loss below 1 on these correct predictions ($r_\theta(x, y_c) > r_\theta(x, y_r)$ or $\Delta r > 0$) while in the case of ($\Delta r=1$, $m=3$), Margin BT gives a relatively high loss (2.1269) even when the prediction is correct - *i.e.* when the model predicted A is slightly better than B when humans annotated A to be much better than B. Such a high loss in Margin BT can cause the model to potentially optimize away from the correct predictions it has made.

| | **Correct Predictions** | | | | **Incorrect Predictions** | | | | Avg. Loss |
|---|---|---|---|---|---|---|---|---|---|
| *Predicted Reward Gap* ($\Delta r$) | $\Delta r=3$ | $\Delta r=3$ | $\Delta r=1$ | $\Delta r=1$ | $\Delta r=-1$ | $\Delta r=-1$ | $\Delta r=-3$ | $\Delta r=-3$ | |
| *Human Annotated Margin* ($m$) | $m=1$ | $m=3$ | $m=1$ | $m=3$ | $m=1$ | $m=3$ | $m=1$ | $m=3$ | |
| Regular Bradley-Terry | 0.0486 | 0.0486 | 0.3133 | 0.3133 | 1.3133 | 1.3133 | 3.0486 | 3.0486 | 1.1800 |
| Margin Bradley-Terry | 0.1269 | 0.6931 | 0.6931 | *2.1269* | 2.1269 | 4.0181 | 4.0181 | 6.0025 | 2.4757 |
| Scaled Bradley-Terry | 0.0486 | 0.1458 | 0.3133 | 0.9399 | 1.3133 | 3.9399 | 3.0486 | *9.1458* | 2.3619 |

Table 10: Loss for various Bradley-Terry loss functions in representative scenarios - $\Delta r$ refers to the gap between chosen and reject rewards as predicted by the model (i.e. $r_\theta(x, y_c) - r_\theta(x, y_r)$) while $m$ refers to the magnitude of preference strength between the chosen and rejected responses as labelled by humans. Correct predictions are made when $r_\theta(x, y_c) > r_\theta(x, y_r)$ or $\Delta r > 0$. Conversely, incorrect predictions are made when $r_\theta(x, y_c) < r_\theta(x, y_r)$ or $\Delta r < 0$.

Compared to Margin BT, Scaled BT also gives a much higher loss of around 50% (9.1458 vs. 6.0025) when drastic mistakes are made ($\Delta r=-3$, $m=3$) - *i.e.* when the model predicted A is much better than B when humans annotated B to be much better than A. It is important to note among our eight representative scenarios, Scaled BT has an average loss similar to Margin BT (2.3619 vs. 2.4767), suggesting that Scaled BT's relative penalties on Incorrect Predictions are heavier.

A side benefit of the Scaled BT loss weighing is that it is likely to be more robust to annotation errors compared to Margin or Regular BT. This is because more human annotation errors are made between slight preference compared to strong preferences, since it's relatively clear when A is much better than B or vice-versa. Therefore, since Scaled BT correspondingly gets higher losses from the "strong preference" samples, it is less affected by annotation errors, which are more commonly found among slight preference samples.

## K    REWARD MODEL ABLATION STUDIES

**Overall**    We use the Scaled Bradley-Terry model initialized with Helpfulness Only SteerLM Regression as a baseline model to compare the effectiveness of various ablations, as this was our best-performing model without model extrapolation. We perform all ablations using similar training settings and hyperparameter search for fairness. Across all ablations in Table 11, no setup exceeds our baseline model suggesting that the design choices we made in training this model are optimal.

| Model Type | Model | Overall | Chat | Chat-Hard | Safety | Reasoning | LR |
|---|---|---|---|---|---|---|---|
| | | | | RewardBench | | | Hyperparams |
| **Baseline (Scaled BT init. with Helpfulness Only SteerLM Regression)** | | 93.7 | 98.0 | 85.7 | 94.3 | 96.7 | 1e-6 |
| **Alt. Data Preprocessing** | including samples with helpfulness spread > 2 | 92.7 | 97.8 | 84.2 | 94.2 | 94.6 | 3e-6 |
| | using all annotations instead of most similar 3 | 92.3 | 98.6 | 82.2 | 93.2 | 95.0 | 1e-6 |
| **Remove overlapping prompts with RewardBench** | | 93.3 | 97.5 | 85.3 | 94.3 | 96.1 | 1e-6 |
| **HelpSteer2 converted into pairwise instead of HelpSteer2-Preference** | | 92.9 | 97.2 | 84.2 | 94.6 | 95.6 | 2e-6 |
| **Helpfulness Only SteerLM Regression (init. with Scaled BT)** | | 92.2 | 98.9 | 84.9 | 91.5 | 93.7 | 1e-6 |

Table 11: Ablation Studies for Reward Models. Higher is better for each category. All models are trained by us using Llama-3.1-70B-Instruct as a base model.

**Alternative Data Preprocessing**   We performed ablation studies on our data preprocessing to better understand the contributions of our data filtering process. We take as baseline for a Reward Model trained on the "smaller, higher-quality dataset" post extensive filtering which reaches 93.7 on RewardBench. Using the same training process as the baseline but without filtering the 10% of samples with wider range (spread >2) results in a degradation of 1 point overall (to 92.7 RewardBench). Using the same training process as the above but using all annotations (up to five) for each task, instead of only the three most similar ones, results in further degradation of 0.4 on Rewardbench to 92.3. These results support our initial belief that that models trained on a smaller, higher-quality dataset perform better than those trained on larger datasets with more noise, suggesting that our data pre-processing to remove noisy samples was suitable.

**Remove overlapping prompts with RewardBench**   Initially, we did not apply a prompt filter to ensure prompts in HelpSteer2-Preference did not overlap with those in RewardBench. At the time of submission (1 Oct 2024), we were not aware of potential prompt overlap between HelpSteer2-Preference and RewardBench. Post-submission, we were informed by RewardBench maintainer that there were 42 prompts in HelpSteer2-Preference that overlap with RewardBench prompts (see details at [3]). In this setting, overlap means that there is at least one identical 7-gram between a HelpSteer2-Preference prompt and any RewardBench prompt. This definition is a low-precision but high-recall definition of overlap (i.e. over-estimate of actual overlaps), because in many of these 42 cases, the 7-gram overlap is due to a similar template used between the prompts. For instance, both prompts can have a common starting template such as "I want you to act as a novelist. ..." but have rather different content after that.

Regardless, this is a negligible portion of both HelpSteer2-Preference (which has 0.42% overlap out of ~10k total prompts) and RewardBench (which has 1.4% overlap out of ~3k total prompts). We believe this small proportion of overlapping prompt is because HelpSteer2-Preference prompts are sourced from ShareGPT, which might have overlapped with the prompt source of constituent datasets of RewardBench as both depend on people voluntarily sharing their ChatGPT conversations.

Out of an abundance of caution, we repeated our training approach for Scaled BT initialized with Helpfulness only SteerLM Regression, which scored 93.7 on RewardBench Overall and the resulting model achieves 93.3 on RewardBench, which is not very different from the original model. Therefore, we are confident that the impact of the extremely limited overlap in prompts is minimal. Please note that even when prompts are overlapping, the model responses are still generated by different models, hence the reward model cannot 'memorize' the reward of a response.

**HelpSteer2 converted into pairwise instead of HelpSteer2-Preference**   To better understand how HelpSteer2-Preference is differentiated from HelpSteer2, we perform an additional RM experiment using HelpSteer2 converted into a pair-wise format. When initialized with Helpfulness-only SteerLM Regression model, training using Scaled BT on the original HelpSteer2 converted into a pair-wise format cannot improve upon the original model (93.0 RewardBench, see Table 1) and achieves 92.9 RewardBench. On the other hand, BT training with HelpSteer2-Preference can improve upon the Helpfulness-only SteerLM Regression model trained with HelpSteer2, reaching 93.7 on RewardBench. This shows that HelpSteer2 and HelpSteer2-Preference contain complementary label information that can be useful in a two-stage RM training process.

**Helpfulness Only SteerLM Regression (init. with Scaled BT)**   When we trained a helpfulness-only SteerLM Regression model initialized with Scaled Bradley Terry model, it has 92.2 Reward-

Bench, which is not better than training a Helpfulness only SteerLM Regression model alone with 93.0 RewardBench. We believe this is because it is not useful to train a Helpfulness-Only SteerLM Regression model initialized on a Bradley-Terry model. The Bradley-Terry Model does not generate rewards within a narrow range and instead can go between -35 and 5 (see Fig. 2). Hence, initializing the regression model from Bradley-Terry model can lead to extremely large losses initially (as initial predictions are between -35 and 5 while ground truth helpfulness ranges from 0 to 4), which might not be better than randomly initializing the regression head.

## L  ALIGNED MODEL ABLATION STUDIES

| Model Type | Data/Reward Model | Aligned Metrics | | | | Hyperparams | |
|---|---|---|---|---|---|---|---|
| | | MT Bench (GPT-4-Turbo) | Mean Response Length (Chars.) | AlpacaEval 2.0 LC (SE) | Arena Hard (95% CI) | LR | KL |
| *DPO* | HelpSteer2-Preference | 8.66 | 1502.2 | 40.4 (1.66) | 52.8 (-2.7, 2.7) | 2e-7 | 0.01 |
| | HelpSteer2 (converted pairwise) | 8.61 | 1609.2 | 32.6 (1.37) | 47.7 (-2.9, 2.6) | 2e-7 | 0.01 |
| *PPO* | Scaled BT + ExPO | 8.74 | 1842.8 | 43.8 (1.76) | 58.6 (-2.9, 2.5) | 1e-6/1e-7 | 0.005/0.01 |
| | Llama3-70B-SteerLM-RM | 8.71 | 1506.2 | 38.2 (1.38) | 49.3 (-2.1, 2.4) | 1e-6 | 0.005 |
| *Starting Model* | Llama-3.1-70B-Instruct | 8.22 | 1728.6 | 38.1 (0.90) | 55.7 (-2.9, 2.7) | - | - |

Table 12: Ablation Studies for Aligned Models. Higher is better for each metric, except Length.

To give a better sense of how other datasets compare to HelpSteer2-Preference terms of RLHF (Section 5), we trained a DPO model using HelpSteer2 dataset (converted into a pairwise setting based on difference in helpfulness), while keeping the training setup and hyperparameter search constant. In addition, we trained a PPO model using Llama3-70B-SteerLM-RM (trained with HelpSteer2) instead of the Scale BT + ExPO reward model (trained with HelpSteer2-Preference), while keeping the training setup and hyperparameter search constant. This model did not benefit from two-stage PPO training, as the one with Scaled BT + ExPO did.

Across both DPO and PPO, HelpSteer2-Preference can be used to align models better than HelpSteer2, in terms of MT Bench, AlpacaEval 2 and Arena Hard. We believe this supports the advantage of collecting a dataset specifically purposed for a pair-wise training setting, over retro-fitting HelpSteer2, which has been collected for a different purpose (single response rating prediction).

## M  FURTHER REINFORCE MODEL METRICS

| Model Type | Model | General Knowledge MMLU (5-Shot, non-CoT) | Math GSM8K (0-shot, non-CoT) | Coding HumanEval (0-shot, non-CoT) |
|---|---|---|---|---|
| *Online RLHF* | REINFORCE | 82.4 | 90.9 | 83.5 |
| *Starting Model* | Llama-3.1-70B-Instruct | 82.2 | 91.6 | 80.5 |

Table 13: Auxiliary Metrics for REINFORCE and Llama-3.1-70B-Instruct

**Overall**  We notice that performance on these benchmarks (MMLU, GSM8K and HumanEval) in Table 13 does not change much after REINFORCE training compared to Llama 3.1 70B Instruct, which we use as a starting policy. This suggests that REINFORCE does not cause catastrophic forgetting of capability gained from prior training.

