# OpenReview forum: "HelpSteer2-Preference: Complementing Ratings with Preferences"
_ICLR.cc/2025/Conference — ICLR 2025 Poster_

### Official Review · Reviewer_wyR3 · 2024-10-29

**Soundness:** 2
**Presentation:** 3
**Contribution:** 3
**Rating:** 6
**Confidence:** 4

**Summary:**

This paper introduces Helpsteer2-preference, a preference dataset that provides additional pairwise preference annotations alongside the existing Likert-5 style rating of the Helpsteer-2 dataset. Using this dataset, the authors conduct head-to-head comparisons between two major types of reward models. Then, they propose a method to combine two types of reward models to achieve better performances on RewardBench and preference learning.

**Strengths:**

- The writing is clear and easy to follow.
- The data collection process is well-designed, with detailed and comprehensive analysis provided.  The proposed dataset provides a direct and fair comparison between Bradley-Terry and Regression-based reward models.
- The proposed method of  combining of different reward models is effective on RewardBench, and improves LLaMA-3.1 through preference alignment.

**Weaknesses:**

- One major concern is the paper/s dependence on HelpSteer2. While the authors states that "converting regression-style data to preference pairs cannot fully capture nuances in preferences", there are no experimental results to support this claim. It would strenghthen the paper to include experimental comparisons with models trained on the preference dataset converted from Helpsteer2. This would help clarify whether the improvement is due to the new preference data itself or the different formulation of reward models.

- Lack of comparison with other reward models or preference dataset in preference alignment (RLHF, DPO).
  - For DPO, what if you apply the converted preference dataset from HelpSteer2?
  - For PPO, can you compare it with other reward models of the same base model(e.g. Llama3-70B-SteerLM-RM, TextEval-Llama3.1-70B)?

**Questions:**

- How does the reward model perform if combined vice-versa, using SteerLM Regression model with the initialization of Bradley-Terry model?
- How many hours of human labors do you use to collect the human annotations?
- How does the proposed reward model combination method compared with model ensembling?

---

> ### Author Response · Authors · 2024-11-21
> **Response to reviewer wyR3 (Part 1 / 2)**
>
> Thank you for your meticulous suggestions and questions!
>
> > One major concern is the paper/s dependence on HelpSteer2. While the authors states that "converting regression-style data to preference pairs cannot fully capture nuances in preferences", there are no experimental results to support this claim. It would strenghthen the paper to include experimental comparisons with models trained on the preference dataset converted from Helpsteer2. This would help clarify whether the improvement is due to the new preference data itself or the different formulation of reward models.
>
> We perform an additional RM experiment using HelpSteer2 converted into a pair-wise format (using similar training setting and hyperparameter search). When initialized with Helpfulness-only SteerLM Regression model, training using Scaled BT on the original Helpsteer2 converted into a pair-wise format cannot improve upon the original model (Overall 93.0 Chat 97.2 Chat-Hard 84.2 Safety 94.6 Reasoning 95.8) and achieves Overall 92.9, Chat 97.2 , Chat-Hard 84.2 , Safety 94.6 , Reasoning 95.6. On the other hand, BT training with HelpSteer2-Preference can improve upon the Helpfulness-only SteerLM Regression model trained with HelpSteer2, reaching 93.7 on RewardBench (Chat 98.0, Chat-Hard 85.7, Safety 94.3 Reasoning 96.7). This shows that HelpSteer2 and HelpSteer2-Preference contain complementary label information that can be useful in a two-stage RM training process.
>
> > Lack of comparison with other reward models or preference dataset in preference alignment (RLHF, DPO).
> For DPO, what if you apply the converted preference dataset from HelpSteer2?
> For PPO, can you compare it with other reward models of the same base model(e.g. Llama3-70B-SteerLM-RM, TextEval-Llama3.1-70B)?
>
> To give a better sense of how other datasets compare in terms of RLHF (Section 5), we trained a DPO model using HelpSteer2 dataset, while keeping the training setup and hyperparameter search constant.
>
> | Dataset used for DPO | MT Bench (GPT-4-Turbo) | Length (Chars.)  | AlpacaEval 2 LC (SE) | Arena Hard (95% CI) |
> |---------------------------------|----------|---------|------------|----------------|
> | HelpSteer2-Preference           | 8.66     | 1502.2  | 40.4 (1.66) | 52.8 (-2.7, 2.7) |
> | HelpSteer2                      | 8.61     | 1609.2  | 32.6 (1.37) | 47.7 (-2.9, 2.6) |
>
>
> In addition, we trained a PPO model using Llama3-70B-SteerLM-RM (trained with HelpSteer2) instead of the Scale BT + ExPO reward model (trained with HelpSteer2-Preference), while keeping the training setup and hyperparameter search constant.
>
> | RM used for PPO (Dataset) | MT Bench (GPT-4-Turbo)| Length (Chars.)  | AlpacaEval 2 LC (SE) | Arena Hard (95% CI) |
> |---------------------------------|----------|---------|------------|----------------|
> | Scale BT + ExPO RM (HelpSteer2-Preference)  | 8.74     | 1842.8  | 43.8 (1.76) |  58.6 (-2.9, 2.5) |
> |  Llama3-70B-SteerLM-RM (HelpSteer2)                  | 8.71     | 1506.2  | 38.2 (1.38) | 49.3  (-2.1, 2.4) |
>
>
> Across both DPO and PPO, HelpSteer2-Preference can be used to align models better than HelpSteer2, in terms of MT Bench, AlpacaEval2 and Arena Hard. We believe this supports the advantage of collecting a dataset specifically purposed for a pair-wise training setting, over retro-fitting HelpSteer2, which has been collected for a different purpose (single response rating prediction).
>
> > How does the reward model perform if combined vice-versa, using SteerLM Regression model with the initialization of Bradley-Terry model?
>
> When we trained a helpfulness-only SteerLM Regression model initialized with Scaled Bradley Terry model, it has 92.2 Reward Bench (Chat 98.9, Chat-Hard 84.9, Safety 91.5, Reasoning 93.7) which is not better than training a Helpfulness only SteerLM Regression model alone with 93.0 RewardBench (Chat 97.2 Chat-Hard 84.2 Safety 94.6 Reasoning 95.8). We believe this is because it is not useful to train a Helpfulness-Only SteerLM Regression model initialized on a Bradley-Terry model. The Bradley-Terry Model does not generate rewards within a narrow range and instead can go between -35 and 5 (based on empirical observation, see Figure 2 of our paper). Hence, initializing the regression model from Bradley-Terry model can lead to extremely large losses initially (as initial predictions are between -35 and 5 while ground truth helpfulness ranges from 0 to 4), which might not be better than randomly initializing the regression head.

---

> ### Author Response · Authors · 2024-11-21
> **Response to reviewer wyR3 (Part 2 / 2)**
>
> > How many hours of human labors do you use to collect the human annotations?
>
> The human annotations took approximately 10,000 hours (+/- 6000 hours). Here, we provide an estimate with a rather large interval since the detailed amount of time across various stages is tracked by our vendor Scale AI and releasing it can constitute sensitive business information. Our estimate is based on per-task time shared by our vendor.
>
> > How does the proposed reward model combination method compared with model ensembling?
>
> When combining the best BT model (Reward Bench 92.7) with the best SteerLM Regression Model (Reward Bench 93.0) using model ensembling (specifically, by averaging the predicted rewards of the two models), the resulting ensemble reaches 92.9 on Reward Bench. More detailedly, it achieves 97.5 on Chat, 84.2 on Chat-Hard, 92.6 on Safety and 97.5 on Reasoning. Therefore, our proposed two-stage training approach (Overall 93.7, Chat 98.0, Chat-Hard, 85.7, Safety: 94.3 Reasoning: 96.7) achieves better performance than doing model ensembling. It is also worth noting that inference with a model ensemble is substantially more resource intensive (i.e. requires more than one reward model to be deployed concurrently), with further engineering challenges in settings like PPO which requires near real-time communication between the various reward models.

---

> > ### Comment · Reviewer_wyR3 · 2024-11-25
> >
> > Thank you for the additional experiments and clarifications. While I still believe that the proposed dataset has certain limitations in its reliance on HelpSteer-2, I acknowledge its value to the community in enabling direct comparisons of different types of reward models and contributing to the advancement of LLM alignment. Therefore, I will maintain my positive score.

---

> > > ### Author Response · Authors · 2024-11-29
> > > **Follow up response to reviewer wyR3**
> > >
> > > Thank you for the quick response. We are unclear about why a “reliance on HelpSteer-2” should be considered as “limitations” of the proposed dataset and would love for the reviewer to elaborate.
> > >
> > > This paper augments an existing dataset (HelpSteer2) to unlock new capabilities (e.g. training a SOTA aligned model measured in terms of Arena Hard) as well as further strengthening previous capabilities (i.e. training a stronger reward model than what can be trained using HelpSteer2 alone, to produce the SOTA reward model measured in terms of RewardBench). We believe that such augmentation characterizes a core aspect of the scientific endeavor, which is to stand on the shoulders of giants, and make meaningful contributions on top of existing works.
> > >
> > > Furthermore, such augmentation was also key to many well-known works. Some examples at the top of our mind are:
> > >
> > > 1. **SQuAD 2.0** [1] (ACL 2018) augments SQuAD [2] by using all original question-answering pairs and adding unanswerable questions. SQuAD 2.0 has been cited over 3000 times.
> > >
> > > 2. **MMLU-Pro** [3] (NeurIPS 2024) augments MMLU [4] by filtering out poor quality questions, adding more questions from a similar distribution and adding more options per question. MMLU-Pro has been used by many including HuggingFace Open LLM Leaderboard [5] and frontier LLM providers such as Anthropic in their recent Claude 3.5 Sonnet release [6].
> > >
> > > Likewise, HelpSteer2-Preference augments HelpSteer2 by adding preference labels (including preference strengths) between the responses to the same prompt as well as an open-field textual reasoning on why a response is preferred over the other. Across the dataset, reward model and aligned models that have been released on HuggingFace, close to half a million downloads have been accumulated, suggesting that the community values the contribution that the HelpSteer2-Preference work brings beyond HelpSteer2.
> > >
> > > More broadly, we are concerned that considering a “reliance” on existing datasets as a “limitation” will discourage subsequent works that aim at making significant improvements to already available resources, to help support the community. This would be a much greater loss to the community beyond the contribution of this paper.
> > >
> > >
> > > [1] https://arxiv.org/abs/1806.03822
> > >
> > > [2] https://arxiv.org/abs/1606.05250
> > >
> > > [3] https://arxiv.org/abs/2406.01574
> > >
> > > [4] https://arxiv.org/abs/2009.03300
> > >
> > > [5] https://huggingface.co/spaces/open-llm-leaderboard/open_llm_leaderboard
> > >
> > > [6] https://www.anthropic.com/claude/sonnet

---

### Official Review · Reviewer_AvcT · 2024-11-03

**Soundness:** 3
**Presentation:** 2
**Contribution:** 2
**Rating:** 5
**Confidence:** 5

**Summary:**

This paper complements the existing HelpSteer2 dataset with preference annotations, showing that pairwise annotation can further distinguish responses with similar likert scores. This dataset, together with its counterpart HelpSteer2, enables a head-to-head comparison between Bradley-Terry and Regression style reward models, by training models with the same prompts and responses in HelpSteer2 and only differing the feedback annotation. Using insights from this comparison, they propose a novel approach combining both methods, achieving superior reqward modeling performances. They also conduct experiments with RL algorithms to demonstrate the effectiveness of the reward models.

**Strengths:**

1. The paper introduces a carefully annotated preference dataset that could be useful. The authors also conduct some analysis on the difference of pointwise and pairwise annotation.

2. This paper compares Bradley-Terry and Regression style reward modeling paradigms, which motivates them to design a new method to make the best of both worlds.

**Weaknesses:**

1. The motivation is somehow overclaimed. In line 013-016, the authors claim that there lacks datasets to enable the comparison between Bradley-Terry and Regression style reward modeling. However, all preference datasets with point-wise scores can be converted into pair-wise setups, which should have been able to allow the comparison between the two paradigms. But I agree with authors that "Regression-style data collection (where responses are rated individually) gives
annotators a different set of expectations for comparative ranking of responses" (line 055-056), so it'd be better to have a pair of dataset directly annotated in point-wise and pair-wise schema. Admittedly, the comparison setup is improved now but it was not totally undoable.

2. The authors conducted a lot of experiments, but it's hard for readers to parse useful information from the experiments with almost no highlights in writings. The abstract mainly focuses on what they did, with no mention on what they have found in their experiments. In the experiment sections, readers would benefit a lot from bolded high-level excerpts to quickly grasp the key findings.

3. The results are good to know, e.g. the comparison between Bradley-Terry and Regression style reward modeling, but personally I don't find it surprising or informative enough to serve as the main claim of a conference paper. It would better to investigate deeper what properties of an algorithm make a good reward model. Also, the experiments are mainly conducted on general chats with subjective metrics (LLM-as-a-judge), thus the conclusions may not be able to generalize to other taks like reasoning, which is known to be very different from chatting. Regarding other experiments, such as section 5, the authors conduct RL with their reward models and datasets mainly for the sake of demonstrating their usefulness. In this case, it seems baselines are needed so that readers can be better aware of the necessity of another preference dataset, given the existence of many good alternatives, and distinguish this dataset with existing ones.

**Questions:**

See weaknesses.

---

> ### Author Response · Authors · 2024-11-21
> **Response to reviewer AvcT (Part 1 / 3)**
>
> Thank you for your thoughtful feedback and suggestions!
>
> > The motivation is somehow overclaimed. In line 013-016, the authors claim that there lacks datasets to enable the comparison between Bradley-Terry and Regression style reward modeling. However, all preference datasets with point-wise scores can be converted into pair-wise setups, which should have been able to allow the comparison between the two paradigms. But I agree with authors that "Regression-style data collection (where responses are rated individually) gives annotators a different set of expectations for comparative ranking of responses" (line 055-056), so it'd be better to have a pair of dataset directly annotated in point-wise and pair-wise schema. Admittedly, the comparison setup is improved now but it was not totally undoable.
>
> We believe that we have not overstated our claim, because the claim is qualified with “However, there is a lack of evidence that either approach is better than the other, **when adequately matched for data**.”  We also further define what we meant by “adequately matched for data” in the introduction (line 46-70), which includes the requirement of data being “Collected for Purpose”.
>
> Nonetheless, we agree with the reviewer that it is indeed possible to convert HelpSteer2 point-wise data into a pair-wise format that supports BT (based on differences in individual helpfulness scores), even though it does not put ‘Bradley-Terry models’ in a fair comparison. To better understand this, we perform additional experiments (using similar settings and hyperparameter search) and discuss them below.
>
> When initialized with Helpfulness-only SteerLM Regression model, training using Scaled BT on the original Helpsteer2 converted into a pair-wise format cannot improve upon the original model (Overall 93.0 Chat 97.2 Chat-Hard 84.2 Safety 94.6 Reasoning 95.8) and achieves Overall 92.9, Chat 97.2 , Chat-Hard 84.2 , Safety 94.6 , Reasoning 95.6. On the other hand, BT training with HelpSteer2-Preference can improve upon the Helpfulness-only SteerLM Regression model trained with HelpSteer2, reaching 93.7 on RewardBench (Chat 98.0, Chat-Hard 85.7, Safety 94.3 Reasoning 96.7). This shows that HelpSteer2 and HelpSteer2-Preference contain complementary label information that can be useful in a two-stage RM training process.
>
> > The authors conducted a lot of experiments, but it's hard for readers to parse useful information from the experiments with almost no highlights in writings. The abstract mainly focuses on what they did, with no mention on what they have found in their experiments. In the experiment sections, readers would benefit a lot from bolded high-level excerpts to quickly grasp the key findings.
>
> We will clarify key results in the abstract and include bolded high-level excerpts in the experiment section to improve readability.

---

> ### Author Response · Authors · 2024-11-21
> **Response to reviewer AvcT (Part 2 / 3)**
>
> > The results are good to know, e.g. the comparison between Bradley-Terry and Regression style reward modeling, but personally I don't find it surprising or informative enough to serve as the main claim of a conference paper. It would better to investigate deeper what properties of an algorithm make a good reward model.
>
> We previously did some investigations into what makes Scaled Bradley-Terry better than Regular and Margin Bradley-Terry, but did not have space to include it into the paper.
>
> To give a clearer justification for Scaled BT, we demonstrate the losses for various BT variants for representative scenarios in the table below. Compared to Regular BT, Scaled BT increases the loss proportionally when the human-annotated margin is larger. Relative to Margin BT, Scaled BT provides a higher loss from incorrect predictions while keeping losses low on correct predictions. Specifically, Scaled BT keeps loss below 1 on these correct predictions ($r_\theta(x, y_{c})$ > $r_\theta(x, y_{r})$ or $\Delta r$ > 0) while in the case of ($\Delta r$=1, $m$=3), Margin BT gives a relatively high loss (2.1269) even when the prediction is correct - i.e. when the model predicted A is slightly better than B when humans annotated A to be much better than B. Such a high loss in Margin BT can cause the model to potentially optimize away from the correct predictions it has made.
>
> |                                       | **Correct Predictions** |        |          |                     | **Incorrect Predictions** |         |           |                      | **Avg. Loss** |
> |--------------------------------------------|-------------------------|--------------|--------------|--------------------------|---------------------------|---------------|---------------|--------------------------|---------------|
> | _Predicted Reward Gap ($\Delta r$)_| $\Delta r$=3            | $\Delta r$=3 | $\Delta r$=1 | $\Delta r$=1             | $\Delta r$=-1             | $\Delta r$=-1 | $\Delta r$=-3 | $\Delta r$=-3            |               |
> | _Human Annotated Margin ($m$)_     | $m$=1                   | $m$=3        | $m$=1        | $m$=3                    | $m$=1                     | $m$=3         | $m$=1         | $m$=3                    |               |
> | Regular Bradley-Terry                      | 0.0486                  | 0.0486       | 0.3133       | 0.3133                   | 1.3133                    | 1.3133        | 3.0486        | 3.0486                   | 1.1800        |
> | Margin Bradley-Terry                       | 0.1269                  | 0.6931       | 0.6931       | **_2.1269_** | 2.1269                    | 4.0181        | 4.0181        | 6.0025                   | 2.4757        |
> | Scaled Bradley-Terry                       | 0.0486                  | 0.1458       | 0.3133       | 0.9399                   | 1.3133                    | 3.9399        | 3.0486        | **_9.1458_** | 2.3619        |
>
>
> Compared to Margin BT, Scaled BT also gives a much higher loss of around 50\% (9.1458 vs. 6.0025) when drastic mistakes are made ($\Delta r$=-3, $m$=3) - i.e. when the model predicted A is much better than B when humans annotated B to be much better than A. It is important to note among our eight representative scenarios, Scaled BT has an average loss similar to Margin BT (2.3619 vs. 2.4767), suggesting that Scaled BT's relative penalties on Incorrect Predictions are heavier.
>
> Moreover, we refer the reviewer to our general comment to all reviewers, where we re-emphasize the significance of our aligned models, which can achieve state of the art results on several alignment benchmarks. Open sourcing our dataset will allow anyone in the community to achieve such results, which we believe is a valuable contribution.

---

> ### Author Response · Authors · 2024-11-21
> **Response to reviewer AvcT (Part 3 / 3)**
>
> > Also, the experiments are mainly conducted on general chats with subjective metrics (LLM-as-a-judge), thus the conclusions may not be able to generalize to other taks like reasoning, which is known to be very different from chatting.
>
> We clarify that LLM-as-a-judge based metrics (MT Bench, Arena Hard and AlpacaEval) are not subjective. Instead, they are highly objective metrics trusted by the model alignment community with public leaderboards, with >50 models. In Table 4, we only selectively included well-known models (such as GPT-4o, Claude 3.5 Sonnet, Llama 3.1 70B/405B) to give a general sense of how our aligned models perform. If the reviewer is interested in the performance of other models, please refer to [1], [2] and [3].
>
> Our evaluation for aligned models includes some reasoning tasks, rather than just “general chats”. Specifically, the prompts for Arena Hard [1] and MT Bench [2] include prompts relating to code, math and other settings requiring logical reasoning (e.g. riddles). For MT Bench, the LLM-provided judgements are further grounded in human-verified reference answers for the categories above, ensuring that the LLM-judgements can account for the correctness of answers.
>
> Finally, we would like to point the reviewer to our general comment to all reviewers, where we discuss evaluation on the LMSYS chatbot arena. Moreover, performance on Chatbot Arena can further be broken down into reasoning-specific categories such as math and coding, where our model performs better than or on par with powerful models such as GPT-4o-2024-08-06 and Llama 3.1 405B.
>
> > Regarding other experiments, such as section 5, the authors conduct RL with their reward models and datasets mainly for the sake of demonstrating their usefulness. In this case, it seems baselines are needed so that readers can be better aware of the necessity of another preference dataset, given the existence of many good alternatives, and distinguish this dataset with existing ones.
>
> To give a better sense of how other datasets compare in terms of RLHF (Section 5), we trained a DPO model using HelpSteer2 dataset, while keeping the training setup and hyperparameter search constant.
>
> | Dataset used for DPO | MT Bench (GPT-4-Turbo) | Length (Chars.)  | AlpacaEval 2 LC (SE) | Arena Hard (95% CI) |
> |---------------------------------|----------|---------|------------|----------------|
> | HelpSteer2-Preference           | 8.66     | 1502.2  | 40.4 (1.66) | 52.8 (-2.7, 2.7) |
> | HelpSteer2                      | 8.61     | 1609.2  | 32.6 (1.37) | 47.7 (-2.9, 2.6) |
>
>
> In addition, we trained a PPO model using Llama3-70B-SteerLM-RM (trained with HelpSteer2) instead of the Scale BT + ExPO reward model (trained with HelpSteer2-Preference), while keeping the training setup and hyperparameter search constant.
>
> | RM used for PPO (Dataset) | MT Bench (GPT-4-Turbo)| Length (Chars.)  | AlpacaEval 2 LC (SE) | Arena Hard (95% CI) |
> |---------------------------------|----------|---------|------------|----------------|
> | Scale BT + ExPO RM (HelpSteer2-Preference)  | 8.74     | 1842.8  | 43.8 (1.76) |  58.6 (-2.9, 2.5) |
> |  Llama3-70B-SteerLM-RM (HelpSteer2)                  | 8.71     | 1506.2  | 38.2 (1.38) | 49.3  (-2.1, 2.4) |
>
>
> Across both DPO and PPO, HelpSteer2-Preference can be used to align models better than HelpSteer2, in terms of MT Bench, AlpacaEval2 and Arena Hard. We believe this supports the advantage of collecting a dataset specifically purposed for a pair-wise training setting, over retro-fitting HelpSteer2, which has been collected for a different purpose (single response rating prediction).
>
> > References
>
> [1] https://github.com/lmarena/arena-hard-auto/tree/main?tab=readme-ov-file#leaderboard
>
> [2] https://tatsu-lab.github.io/alpaca_eval/
>
> [3] https://github.com/lm-sys/FastChat/pull/3158

---

> ### Comment · Reviewer_AvcT · 2024-12-03
> **Respond to Author Responses**
>
> Thanks for your detailed response.
>
> Regarding the dataset contribution, as far as I can see from your response (1/3), HelpSteer2-Preference leads to marginal improvement compared to HelpSteer2 pointwise. Thus, it seems that it's good for the community to have such dataset, but it is not a critical adds-on that bridges an important gap. I respectfully acknowledge authors' efforts on making this dataset, but I suppose that there are good alternatives and we may also be able to produce strong models by properly mixing them, unless proven wrong.
>
> Secondly, regarding the supplemented BT experiments, do you have any takeaway for "what properties of an algorithm make a good reward model"? Maybe I did not get it correctly, but these findings only tell us how to choose the three BT variants in practice. But are different kinds of data requires different algorithms? Besides BT, what else can we try and why? For example, lots of recent work found that BT leads to the decrease in rewards of both chosen data and rejected data, does it apply to reward modeling? If so, how to improve them based on empirical observations?
>
> Finally, I may not agree that LLM-as-a-judge is a subjective metric. LLMs are known to have biases, e.g. style, length, etc. I trust LMSYS Arena more, yet not 100%, because according to my own scan many prompts there are shallow and may not really reflect model ability.
>
> In general, I think this work is well executed, and is rigorous, but I still believe authors can find more in-depth insights than current versions.

---

> > ### Author Response · Authors · 2024-12-03
> > **Follow up response to reviewer AvcT (Part 2 / 2)**
> >
> > > Finally, I may not agree that LLM-as-a-judge is a subjective metric. LLMs are known to have biases, e.g. style, length, etc. I trust LMSYS Arena more, yet not 100%, because according to my own scan many prompts there are shallow and may not really reflect model ability.
> >
> >
> > LLMs do indeed have biases relating to e.g. style and length. In the paper, we report metrics that were best available at time of submission to mitigate such biases. For instance, when choosing to use AlpacaEval 2, we used the Length-Corrected (LC) version which penalizes models for having overly-long responses and thereby counteract length bias.
> >
> >
> > LMSYS Arena is a popular voting platform with millions of anonymous votes and has been trusted by companies such as OpenAI, Anthropic and Meta to test for human preferences of responses generated by their models. **While it is certainly not perfect, LMSYS Arena together with AlpacaEval 2 LC, Arena Hard and MT Bench form the best practices of evaluating aligned models for human preference** [1], [2], [3]. We note that there are some other evaluations (e.g. MATH, MMLU and HumanEval), which we measured for models in our response to reviewer yVas. However, these evaluations do not measure human preference of responses, which is our goal for training aligned models.
> >
> >
> > > In general, I think this work is well executed, and is rigorous, but I still believe authors can find more in-depth insights than current versions.
> >
> >
> > Thank you for your appreciation on the well-execution and rigorousness of our work and we hope we have clarified the reviewer’s remaining concerns.
> >
> > > References
> >
> >
> > [1] https://arxiv.org/abs/2405.14734 SimPO: Simple Preference Optimization with a Reference-Free Reward
> >
> > [2] https://arxiv.org/abs/2407.19594 Meta-Rewarding Language Models: Self-Improving Alignment with LLM-as-a-Meta-Judge
> >
> > [3] https://arxiv.org/abs/2405.07863 RLHF workflow: From reward modeling to online RLHF

---

> ### Author Response · Authors · 2024-12-03
> **Follow up response to reviewer AvcT (Part 1 / 2)**
>
> > Regarding the dataset contribution, as far as I can see from your response (1/3), HelpSteer2-Preference leads to marginal improvement compared to HelpSteer2 pointwise. Thus, it seems that it's good for the community to have such dataset, but it is not a critical adds-on that bridges an important gap.
>
>
> In terms of Reward Bench numbers from our Response 1 / 3,  the absolute difference can seem small - 92.9 RewardBench when training Scaled BT on the original Helpsteer2:  Chat 97.2 , Chat-Hard 84.2 , Safety 94.6 , Reasoning 95.6) vs 93.7 on RewardBench when training Scaled BT on the Helpsteer2-Preference (Chat 98.0, Chat-Hard 85.7, Safety 94.3 Reasoning 96.7). However, it’s important to compare these two numbers in terms of the absolute errors they make - which are 7.1% (100 - 92.9)  against 6.3% (100 - 93.7) respectively. Looking at their absolute errors, we see that the **training on HelpSteer2-Preference instead of HelpSteer2 leads to a substantial 11.3% reduction in relative errors**  ((7.1 - 6.3) / 7.1). The importance of focusing on this relative error reduction can also be demonstrated by there being 6 models in between 92.9 and 93.7 on Reward Bench, meaning that that at time of submission, the HelpSteer2-Preference trained model (93.7 Reward Bench) would be No. 3 while the HelpSteer2 trained model (92.9 Reward Bench) would be No. 9. As a note, we also trained the No. 1 model (94.1 Reward Bench) using ExPO on top of the HelpSteer2-Preference trained model. Therefore, it cannot be said that the improvement for HelpSteer2-Preference over HelpSteer2 on training reward models is marginal.
>
>
> Furthermore, our Response 3 / 3 highlights that HelpSteer2-Preference is capable of producing much stronger aligned models using either DPO or PPO. Specifically, when training DPO model, HelpSteer2 trained model achieves 47.7 on Arena Hard while HelpSteer2-Preference trained model achieves 52.8 on Arena Hard. For PPO, HelpSteer2 trained model achieves 49.3 on Arena Hard while HelpSteer2-Preference trained model achieves 58.6. **To give a better sense of the benefit of using HelpSteer2-Preference over HelpSteer2 (5.1 for DPO and 9.3 for PPO on Arena Hard), the scores on Arena Hard leaderboard for claude-3-haiku-20240307 (41.5) and claude-3-sonnet-20240229 (46.8) differ by 5.3**. Therefore, aligned models can be substantially improved by training on HelpSteer2-Preference instead of HelpSteer2.
>
>
> > I respectfully acknowledge authors' efforts on making this dataset, but I suppose that there are good alternatives and we may also be able to produce strong models by properly mixing them, unless proven wrong.
>
>
> At time of submission, we trained the No. 1 reward model (measured using RewardBench’s public leaderboard with more than 140+ models) as well as the No. 1 aligned model (measured using Arena Hard’s public leaderboard). It might indeed be possible to train strong models by mixing other datasets, but **no single published work has been able to train models as strong as ours (in terms of RewardBench and Arena Hard) using alternative/mixed datasets at time of submission**. Another factor to consider is that our proposed dataset (HelpSteer2-Preference) has an enterprise-friendly license (CC-BY-4.0), meaning that commercial use is permitted while many alternative datasets are limited to academic/non-commercial use.
>
> > Secondly, regarding the supplemented BT experiments, do you have any takeaway for "what properties of an algorithm make a good reward model"? Maybe I did not get it correctly, but these findings only tell us how to choose the three BT variants in practice. But are different kinds of data requires different algorithms? Besides BT, what else can we try and why? For example, lots of recent work found that BT leads to the decrease in rewards of both chosen data and rejected data, does it apply to reward modeling? If so, how to improve them based on empirical observations?
>
>
> We agree that these additional research questions raised by the reviewer are indeed interesting research directions. However, the main objective of the paper lies in introducing a new dataset (HelpSteer2-Preference) and optimally using this dataset to train SoTA reward and aligned models. **Writing a paper with a focus different from algorithmic analysis should not be a weakness for a paper submitted to ICLR with a primary area indicated as “Datasets and Benchmarks”.**

---

### Official Review · Reviewer_KjD8 · 2024-11-03

**Soundness:** 3
**Presentation:** 3
**Contribution:** 3
**Rating:** 6
**Confidence:** 4

**Summary:**

This work addresses two major paradigms: Bradley-Terry (BT) and Regression style models, examining which is better when matched for data. The main challenge in previous studies was the lack of comparable datasets collected using both methodologies. To resolve this, the authors introduce the HelpSteer2-Preference dataset, which complements existing ratings with preference annotations, including human-written justifications. This dual-annotation allows a head-to-head comparison between BT and Regression models under controlled data conditions.

**Strengths:**

First, it complements existing ratings with detailed preference annotations, allowing for more nuanced training of reward models. By collecting preference data alongside Likert-scale ratings, the dataset enables direct comparisons between models trained under different paradigms (e.g., Bradley-Terry vs. Regression), which was previously difficult due to incompatible data formats in existing datasets. This dual approach provides a richer training ground for testing and developing models that need to evaluate comparative quality.

Second, the dataset includes human-written justifications for preferences, adding a layer of interpretability. This is particularly valuable for training models that can both assess and explain their choices, supporting applications where transparency is important, such as in human-in-the-loop systems. The justifications make the dataset unique by facilitating the development of models that are not just performance-driven but also capable of detailed decision-making explanations.

Third, the dataset maintains high-quality standards, demonstrated by its filtering processes and the steps taken to ensure strong inter-annotator agreement. By curating data that passes rigorous quality checks, the authors improve the signal-to-noise ratio, making it more effective for training robust models. This attention to quality addresses common issues in training data, where noisy or inconsistent annotations can limit model performance.

Overall, the HelpSteer2-Preference dataset's strengths lie in its dual-format data collection, the inclusion of human justifications for enhanced interpretability, and its commitment to high-quality, reliable annotations. This makes it a valuable resource for advancing research in reward modeling and instruction-following alignment in language models.

**Weaknesses:**

While the HelpSteer2-Preference dataset brings several innovations and benefits, it also has certain weaknesses that could impact its applicability and use in research and development:

First, the dataset's format, which includes both ratings and preferences, may require more complex preprocessing and handling by researchers. The dual data structure means that models need to be capable of leveraging both individual ratings and comparative preferences, which could increase the implementation complexity for teams that are not equipped with extensive data processing expertise.

Second, despite the inclusion of human-written justifications enhancing interpretability, these justifications may introduce variability due to subjective writing styles or annotator biases. This subjectivity can make it challenging to maintain consistency across annotations and could potentially affect the model's ability to generalize when trained with such data. Additionally, training on justifications may require more advanced model architectures that can process and utilize free-form text, which may not be available or practical for all research teams.

Third, while the dataset emphasizes high-quality annotations with steps taken to filter out low-confidence or inconsistent data, this strict filtering reduces the overall volume of usable data. The exclusion of samples with high variability among annotators may result in a smaller training set, potentially limiting model performance, especially for models that benefit from a large and diverse dataset. This trade-off between quality and quantity could be a bottleneck for researchers who need extensive data for training large-scale models.

Lastly, the HelpSteer2-Preference dataset is tailored to specific types of model training and evaluations, such as Bradley-Terry and Regression-based reward models. This specialization may limit its generalizability to other types of machine learning tasks or applications that do not align with these paradigms. Researchers working on models with different objectives or those requiring more diverse data formats might find the dataset less adaptable.

**Questions:**

1. How were the annotators trained to provide consistent ratings and justifications, and what measures were in place to minimize subjective bias?
2. How well does the dataset generalize to real-world applications beyond reward modeling and instruction-following?
3. What is the impact of filtering out samples with high annotation variability on the overall representativeness and diversity of the dataset?
4. Would models trained on a smaller, higher-quality dataset perform better than those trained on larger datasets with more noise?
5. How might the dataset be expanded or adapted to cover other dimensions of reward modeling, such as domain-specific use cases or more complex types of preferences?

---

> ### Author Response · Authors · 2024-11-21
> **Response to reviewer KjD8 (Part 1 / 2)**
>
> Thank you for your comments! We would like to first address the main weaknesses highlighted in your review:
>
> > First, the dataset's format, which includes both ratings and preferences, may require more complex preprocessing and handling by researchers. The dual data structure means that models need to be capable of leveraging both individual ratings and comparative preferences, which could increase the implementation complexity for teams that are not equipped with extensive data processing expertise.
>
> We do not believe this requires “extensive data processing expertise” since this is a simple combination of two commonly used formats. In addition, models trained on this data do not necessarily “need to be capable of leveraging both individual ratings and comparative preferences” – in fact, the models we train in this paper only use either individual ratings *or* comparative preferences during a single training run, not both.
>
> > Second, despite the inclusion of human-written justifications enhancing interpretability, these justifications may introduce variability due to subjective writing styles or annotator biases. This subjectivity can make it challenging to maintain consistency across annotations and could potentially affect the model's ability to generalize when trained with such data. Additionally, training on justifications may require more advanced model architectures that can process and utilize free-form text, which may not be available or practical for all research teams.
>
> It is true that free-form text can indeed have a lot of variance in their presentation, and it is not obvious how to best leverage such justifications. However, we believe that releasing them may support further research towards taking advantage of human-written explanations to train stronger AI evaluators, and thus consider them to be of significant potential value in spite of these challenges. We also politely disagree that “training on justifications require more advanced model architectures that can process and utilize free-form text”, since any commonly used model architecture (Transformer, RNNs, SSMs) that can generate text can be directly used as-is.
>
> > Third, while the dataset emphasizes high-quality annotations with steps taken to filter out low-confidence or inconsistent data, this strict filtering reduces the overall volume of usable data. The exclusion of samples with high variability among annotators may result in a smaller training set, potentially limiting model performance, especially for models that benefit from a large and diverse dataset. This trade-off between quality and quantity could be a bottleneck for researchers who need extensive data for training large-scale models.
>
> Although this indeed reduces the size of the overall dataset, we have found it to be beneficial to focus on quality rather than quantity based on follow-up experiments (see more details in response to Question 4 below).
>
> > Lastly, the HelpSteer2-Preference dataset is tailored to specific types of model training and evaluations, such as Bradley-Terry and Regression-based reward models. This specialization may limit its generalizability to other types of machine learning tasks or applications that do not align with these paradigms. Researchers working on models with different objectives or those requiring more diverse data formats might find the dataset less adaptable.
>
> Our dataset is indeed specialized towards those types of reward models, which to our knowledge are the most commonly used in the literature [1], [2], [3]. If the reviewer has examples of “other types of machine learning tasks or applications that do not align with these paradigms”, relevant to reward modeling, we would be happy to add a discussion about those in the paper.
>
> **Regarding your questions:**
>
> > 1. How were the annotators trained to provide consistent ratings and justifications, and what measures were in place to minimize subjective bias?
>
>
> Annotators were provided with detailed guidelines that are designed to minimize subjective bias, by giving specific guidance on how to assign ratings and preferences (see e.g. Appendix B for the preference ranking guidelines). A large part of the responsibility to ensure consistency in annotations lies with our vendor (Scale AI), who implemented their own internal training process, automated checks and human review pipeline, the details of which cannot be provided due to the business sensitive nature of this information. Some of the authors also regularly reviewed a small amount of annotations per week throughout the data collection process, to provide feedback to the vendor on common mistakes and inconsistencies they would notice, and clarify edge cases. Finally, we also asked 3-5 annotators to label the same task, removed outliers and then took their average, which minimizes the occurrence of biases from individual annotators being reflected in the resulting dataset (see details in lines 121-131).

---

> ### Author Response · Authors · 2024-11-21
> **Response to reviewer KjD8 (Part 2 / 2)**
>
> > 2. How well does the dataset generalize to real-world applications beyond reward modeling and instruction-following?
>
> This dataset is specifically designed to be used for domain-general reward modeling and general instruction following, and as a result we do not expect it to generalize to other kinds of applications. If the reviewer has specific examples in mind of such real-world applications, please let us know so that we can either investigate it empirically or add some discussion about it.
>
> > 3. What is the impact of filtering out samples with high annotation variability on the overall representativeness and diversity of the dataset?
>
> See response to Question 4 below, as we believe that the best way to quantify the “representativeness and diversity” of the dataset is to evaluate the quality of reward models trained on it (but please let us know if you have other metrics to suggest).
>
> > 4. Would models trained on a smaller, higher-quality dataset perform better than those trained on larger datasets with more noise?
>
> We agree this is an important question to investigate, and have added additional empirical results to justify our data filtering process:
> We take as baseline for a Reward Model trained on the “smaller, higher-quality dataset” the Scaled Bradley-Terry model from our submission, initialized with the Helpfulness-only Regression model (RewardBench scores: Overall 93.7, Chat 98.0, Chat-Hard, 85.7, Safety: 94.3 Reasoning: 96.7)
> Using the same training process as (a) but without filtering the 10% of samples with wider range (spread >2) results in a degradation of 1 point overall (Overall 92.7, Chat 97.8, Chat-Hard: 84.2, Safety: 94.2, Reasoning 94.6)
> Using the same training process as (b) but using all annotations (up to five) for each task, instead of only the three most similar ones, results in further degradation of 0.4 (Overall 92.3, Chat 98.6, Chat-Hard 82.2, Safety 93.2, Reasoning 95.0)
>
> These results indeed confirm that “models trained on a smaller, higher-quality dataset perform better than those trained on larger datasets with more noise”.
>
> > 5. How might the dataset be expanded or adapted to cover other dimensions of reward modeling, such as domain-specific use cases or more complex types of preferences?
>
> This is a rather broad question, to which we can provide a few starting points below:
>
> 1. We tried to make our guidelines generic enough that we believe they should, for the most part, apply to many domain-specific use cases. However, it is likely that some domains may benefit from custom changes: for instance, to evaluate generated code it could be useful to explicitly ask annotators to indicate if the code contains enough comments, and whether those comments are relevant. Our recommendation would be to start annotating a small amount of new domain-specific tasks with the current guidelines, and expand them based on gaps identified by annotators during this process.
>
> 2. One example of “more complex types of preferences” could be ranking more than two responses. Annotation effort typically increases quadratically with the number of responses for preference ranking, but our dual annotation process (ratings + rankings) should help alleviate this effort. Indeed, each response can first be rated individually (linear scaling), after which a partial ranking may be efficiently obtained from those ratings, with only responses with equal (or very close) ratings requiring a more in-depth comparison.
>
> 3. Another example of “more complex types of preferences” could be to introduce preferences across multiple more fine-grained axes (ex: “which response best follows all constraints in the prompt?”, “which response has the best formatting?”, “which response is the safest?”, etc.). Additional ratings could be associated with each such axes so as to keep the benefits of the dual “rating + ranking” data structure. This would require expanding the guidelines to explain how to rate and rank responses along those new axes, along with specific examples. To the best of our knowledge, how to use such more fine-grained preference data (to train better reward models and/or better aligned models) remains an open research question (ex: [4]) that is an interesting avenue for future work.
>
> > References
>
> [1] https://arxiv.org/abs/2403.13787 RewardBench: Evaluating Reward Models for Language Modeling
>
> [2] https://openreview.net/pdf?id=GqDntYTTbk Starling-7B: Improving Helpfulness and Harmlessness with RLAIF
>
> [3] https://arxiv.org/abs/2406.12845 Interpretable Preferences via Multi-Objective Reward Modeling and Mixture-of-Experts
>
> [4] https://arxiv.org/abs/2306.01693 Fine-grained human feedback gives better rewards for language model training

---

### Official Review · Reviewer_yVaS · 2024-11-04

**Soundness:** 4
**Presentation:** 3
**Contribution:** 3
**Rating:** 6
**Confidence:** 4

**Summary:**

This paper presents HelpSteer2, a dataset designed to facilitate research in RM for aligning language models with human preferences. This work addresses the gap of fairly comparison of BT and regression style of RM by releasing preference annotations compatible with the BT style alongside existing ratings formatted for Regression-style training. The authors then conduct a head-to-head comparison of the BT and Regression models when matched for data quality, providing valuable insights into the effectiveness of each approach. They also propose a hybrid model that combines elements of both BT and Regression approaches, achieving state-of-the-art performance on the RewardBench benchmark. Additionally, the authors open-source the HelpSteer2 dataset and trained reward model to promote further research in RM alignment tasks.

**Strengths:**

- The authors have created and open-sourced the HelpSteer2 dataset, which includes high-quality preference annotations and justifications specifically designed for reward model (RM) training.
- The paper conducts a detailed, head-to-head comparison between Bradley-Terry (BT) and Regression-style reward models, providing insights into the relative strengths and weaknesses of each approach.

**Weaknesses:**

- While the hybrid approach is innovative, it seems to be an incremental improvement rather than a fundamentally new modeling technique.
- Lack of statistical significance testing to validate the effectiveness of proposed hybrid modeling method.
- The paper does not evaluate the aligned models on standard language modeling tasks, which would help in quantifying the alignment tax.

**Questions:**

- Could the authors provide statistical significance testing to validate the reported performance improvements of the hybrid model? This would strengthen the robustness of the conclusions.
- Have the authors considered evaluating the align models on standard language modeling tasks to assess the alignment tax?
- Based on the appendix, it appears that the RL training involved approximately 10k examples, and the model was trained for fewer than 100 iterations. Could the authors clarify the stopping criteria for the training process? Additionally, could you specify the KL divergence value of the choosen checkpoints from the reference policy? This information would provide better insight into the training stability and convergence behavior of the reward model.
- Are the conclusions valid for cases where, e.g, 5 responses ranking are used for RM training?

---

> ### Author Response · Authors · 2024-11-21
> **Response to reviewer yVas (Part 1 / 2)**
>
> Thank you for your helpful questions and suggestions!
>
> > While the hybrid approach is innovative, it seems to be an incremental improvement rather than a fundamentally new modeling technique.
>
> We clarify that we propose a hybrid approach combining both SteerLM Regression and a novel variant of Bradley-Terry (Scaled Bradley-Terry). Specifically, our paper introduces a simple change to the Bradley-Terry modeling approach, which we show to be empirically useful and can be easily adopted by others (with only a few lines of code changes). We believe that such pragmatic innovation should not be penalized, especially considering that this allowed us to train the strongest Reward Model on Reward Bench (out of 140+ models) at time of submission. On the other hand, many Reward models trained with ‘fundamentally new modeling techniques’ [2], [4], [8] perform poorly relative to our Reward Model.  As a community, we should be able to celebrate simple but highly effective improvements alongside fundamentally new techniques, instead of solely focussing on one over the other.
>
> To better highlight the contribution of Scaled Bradley-Terry, we did further theoretical investigations into what makes Scaled Bradley-Terry better than Regular and Margin Bradley-Terry. We demonstrate the losses for various BT variants for representative scenarios in the table below to clearly illustrate the advantages of Scaled BT. Compared to Regular BT, Scaled BT increases the loss proportionally when the human-annotated margin is larger. Relative to Margin BT, Scaled BT provides a higher loss from incorrect predictions while keeping losses low on correct predictions. Specifically, Scaled BT keeps loss below 1 on these correct predictions ($r_\theta(x, y_{c})$ > $r_\theta(x, y_{r})$ or $\Delta r$ > 0) while in the case of ($\Delta r$=1, $m$=3), Margin BT gives a relatively high loss (2.1269) even when the prediction is correct - i.e. when the model predicted A is slightly better than B when humans annotated A to be much better than B. Such a high loss in Margin BT can cause the model to potentially optimize away from the correct predictions it has made.
>
> |                                       | **Correct Predictions** |        |          |                     | **Incorrect Predictions** |         |           |                      | **Avg. Loss** |
> |--------------------------------------------|-------------------------|--------------|--------------|--------------------------|---------------------------|---------------|---------------|--------------------------|---------------|
> | _Predicted Reward Gap ($\Delta r$)_| $\Delta r$=3            | $\Delta r$=3 | $\Delta r$=1 | $\Delta r$=1             | $\Delta r$=-1             | $\Delta r$=-1 | $\Delta r$=-3 | $\Delta r$=-3            |               |
> | _Human Annotated Margin ($m$)_     | $m$=1                   | $m$=3        | $m$=1        | $m$=3                    | $m$=1                     | $m$=3         | $m$=1         | $m$=3                    |               |
> | Regular Bradley-Terry                      | 0.0486                  | 0.0486       | 0.3133       | 0.3133                   | 1.3133                    | 1.3133        | 3.0486        | 3.0486                   | 1.1800        |
> | Margin Bradley-Terry                       | 0.1269                  | 0.6931       | 0.6931       | **_2.1269_** | 2.1269                    | 4.0181        | 4.0181        | 6.0025                   | 2.4757        |
> | Scaled Bradley-Terry                       | 0.0486                  | 0.1458       | 0.3133       | 0.9399                   | 1.3133                    | 3.9399        | 3.0486        | **_9.1458_** | 2.3619        |
>
>
> Compared to Margin BT, Scaled BT also gives a much higher loss of around 50\% (9.1458 vs. 6.0025) when drastic mistakes are made ($\Delta r$=-3, $m$=3) - i.e. when the model predicted A is much better than B when humans annotated B to be much better than A. It is important to note among our eight representative scenarios, Scaled BT has an average loss similar to Margin BT (2.3619 vs. 2.4767), suggesting that Scaled BT's relative penalties on Incorrect Predictions are heavier.
>
> Moreover, we refer the reviewer to our general comment to all reviewers, where we re-emphasize the significance of our aligned models, which can achieve state of the art results on several alignment benchmarks.

---

> ### Author Response · Authors · 2024-11-21
> **Response to reviewer yVas (Part 2 / 2)**
>
> > Lack of statistical significance testing to validate the effectiveness of proposed hybrid modeling method. Could the authors provide statistical significance testing to validate the reported performance improvements of the hybrid model? This would strengthen the robustness of the conclusions.
>
> We follow best practices in reporting statistical significance. However, Reward Bench [1] - the most trusted leaderboard for reward models (with more than 140+ models on the leaderboard at time of submission) doesn’t come with such statistics, which means that all papers [2], [3] and [4] that report it do not report statistical significance. On the other hand, when metrics come with some representation of statistical significance (e.g. Standard Error for AlpacaEval 2.0 Length-Corrected and 95% Confidence Interval for Arena-Hard), we do report them, as seen in Table 4.
>
> > The paper does not evaluate the aligned models on standard language modeling tasks, which would help in quantifying the alignment tax. Have the authors considered evaluating the align models on standard language modeling tasks to assess the alignment tax?
>
> Thank you for the suggestion. Below, we present a table with other language modeling benchmarks, including MMLU, GSM8K, and HumanEval, measured without the use of Chain-of-Thought (CoT). Overall, we notice that performance on these benchmarks does not change much after REINFORCE training compared to Llama 3.1 70B Instruct, which we use as a starting policy.
>
> | Model                         | MMLU (General Knowledge, 5-Shot) | GSM8K (Math, 0-shot) | HumanEval (Coding, 0-shot) |
> |-------------------------------|----------------------------------|-----------------------|----------------------------|
> | REINFORCE                      | 82.4                             | 90.9                  | 83.5                       |
> | Llama 3.1 70B Instruct         | 82.2                             | 91.6                  | 80.5                       |
>
>
>
> > Based on the appendix, it appears that the RL training involved approximately 10k examples, and the model was trained for fewer than 100 iterations. Could the authors clarify the stopping criteria for the training process? Additionally, could you specify the KL divergence value of the choosen checkpoints from the reference policy? This information would provide better insight into the training stability and convergence behavior of the reward model.
>
> We evaluate the performance of the trained model on MT-Bench every 5 steps during training, and find that this score plateaus in around 105 steps (=1 epoch), and hence stopped our training. The KL divergence value of the chosen checkpoint is roughly 0.2. We remark that in practice, we find that a lower KL penalty beta performs better after trying both 0.01 and 0.1. In general, we find training with REINFORCE to be quite stable (see Figure 3).
>
> > Are the conclusions valid for cases where, e.g, 5 responses ranking are used for RM training?
>
> In datasets with more than two responses per prompt such as Nectar [5] or Ultrafeedback [6], the baseline reward modeling approach is commonly based on pairwise Bradley Terry, achieved by decomposing K-responses into K(K − 1)/2 pairs. We did not experiment with techniques such as k-wise comparisons [7] since the HelpSteer2-Preference data only contains 2 responses per prompt. Hence, we cannot claim that our conclusions apply to such cases.
>
> > References
>
> [1] https://arxiv.org/abs/2403.13787 RewardBench: Evaluating Reward Models for Language Modeling
>
> [2] https://arxiv.org/abs/2409.14664 Direct Judgement Preference Optimization
>
> [3] https://arxiv.org/abs/2406.12845 Interpretable Preferences via Multi-Objective Reward Modeling and Mixture-of-Experts
>
> [4] https://arxiv.org/abs/2408.02666 Self-Taught Evaluators
>
> [5] https://openreview.net/pdf?id=GqDntYTTbk Starling-7B: Improving Helpfulness and Harmlessness with RLAIF
>
> [6] https://arxiv.org/abs/2310.01377 UltraFeedback: Boosting Language Models with Scaled AI Feedback
>
> [7] https://arxiv.org/abs/2301.11270 Principled Reinforcement Learning with Human Feedback from Pairwise or K-wise Comparisons
>
> [8] https://arxiv.org/abs/2408.11791 Critique-out-Loud Reward Models

---

> > ### Comment · Reviewer_yVaS · 2024-11-29
> >
> > Thank you to the authors for providing the detailed explanation.
> >
> > I will maintain my positive assessment and adjust the soundness rating to better reflect the clarifications provided.

---

> ### Author Response · Authors · 2024-11-29
> **Follow up comment to Reviewer yVaS**
>
> Thank you for your response! We see that the reviewer has rated the work highly on the individual dimensions (Soundness: 4: excellent; Presentation: 3: good ; Contribution: 3: good) but the overall Rating (6: marginally above the acceptance threshold) seems somewhat incongruent with the individual dimensions.
>
> Were any of the weaknesses and questions highlighted in the reviewer's original review not addressed satisfactorily by our earlier responses? If so, please let us know and we will be happy to provide additional clarifications, experiments and/or analysis to address them.

---

### Official Review · Reviewer_eXue · 2024-11-04

**Soundness:** 3
**Presentation:** 3
**Contribution:** 3
**Rating:** 6
**Confidence:** 4

**Summary:**

This paper explores whether a Bradley-Terry or regression approach is more effective for reward modeling and proposes a novel method that combines both styles. The conclusion is intriguing: the format in which data is collected and the specific model training approach do not significantly impact the results.

**Strengths:**

This paper provides an excellent dataset that will be highly valuable to the community. It addresses an important question: whether the Bradley-Terry style or regression style is more effective for reward modeling. Additionally, the reward model (RM) proposed here is capable of effectively training a LLaMA-70B model.

**Weaknesses:**

I'm glad to see that the BT training method with an added margin (eq.2) is effective. However, this finding somewhat conflicts with the results in the Llama 3 paper, where the authors mention: "The training objective is the same as Llama 2, except that we removed the margin term in the loss, as we observed diminishing improvements after data scaling." What could be the reason behind this discrepancy?

I understand that this work provides an excellent dataset and rigorous experimental validation. However, I would like to see more analysis, such as:

1. A comparison of results on out-of-distribution prompts.

2. A consideration of annotation costs: since preference annotations are likely cheaper and easier to scale than Likert scores, a fair comparison might require factoring in annotation costs rather than simply matching dataset sizes.

3. An analysis of robustness to annotation errors. For instance, assume that preference annotations have a 5% chance of being reversed, or that Likert scores have a similar error rate (where a 9 might be incorrectly marked as a 1, rather than just slightly off, like 8). Which reward modeling approach is more robust to such errors?

These aspects would provide valuable insights into the practical implications of each approach.

**Questions:**

Has a prompt filter been applied to ensure that prompts from the rewardbench are not included in the training set?

Additionally, how does SteerLM regression, initialized with the Bradley-Terry model, perform?

---

> ### Author Response · Authors · 2024-11-21
> **Response to reviewer eXue (Part 1 / 3)**
>
> Thank you for your insightful comments!
>
> > I'm glad to see that the BT training method with an added margin (eq.2) is effective. However, this finding somewhat conflicts with the results in the Llama 3 paper, where the authors mention: "The training objective is the same as Llama 2, except that we removed the margin term in the loss, as we observed diminishing improvements after data scaling." What could be the reason behind this discrepancy?
>
> We clarify that there is no conflict between the results of our paper and the results shown in the Llama 3 paper [1].  Specifically, we find that Margin BT (eq. 2)  is not more effective than Regular BT (eq. 1) since both achieve 91.5 on Overall RewardBench, as shown in Table 1. We introduce Scaled BT (eq. 3), which is a distinct approach from Margin BT. See line 245 for more information.
>
>
>
> > A comparison of results on out-of-distribution prompts.
>
> The RewardBench result that we show in Table 1 contains both in-distribution (Chat/Chat-Hard) and out-of-distribution prompts (Safety/Reasoning). Chat/Chat-Hard are in-domain because our prompts were sourced from ShareGPT to compare the helpfulness of responses in these general-domain chat settings (i.e. how people might use LLMs through an UI). Safety and Reasoning are out-of-domain because we did not include safety-related or code-related prompts in this data collection.
>
> > A consideration of annotation costs: since preference annotations are likely cheaper and easier to scale than Likert scores, a fair comparison might require factoring in annotation costs rather than simply matching dataset sizes.
>
> When there are thorough guidelines that require annotators to detailedly and fully inspect the responses (as in this data collection), the bulk of the time is spent on reading, examining and critiquing the responses. A very small fraction of time is spent on the act of indicating Likert/Preference. Therefore, while indicating a preference is technically faster than Likert scores, the total amount of time that annotators typically spend on a task in a Likert and Preference style are very similar. Hence, controlling for dataset size alone is a good proxy for matching annotation time and hence annotation cost.

---

> ### Author Response · Authors · 2024-11-21
> **Response to reviewer eXue (Part 2 / 3)**
>
> > An analysis of robustness to annotation errors. For instance, assume that preference annotations have a 5% chance of being reversed, or that Likert scores have a similar error rate (where a 9 might be incorrectly marked as a 1, rather than just slightly off, like 8). Which reward modeling approach is more robust to such errors?
>
> There are two ways in which we can interpret the reviewer’s question and we address each way in order.
>
> **1. Is Scaled Bradley-Terry reward-modeling more robust to annotation errors than Margin or Regular Bradley-Terry?**
>
> We believe the Scaled BT model is likely to be more robust to annotation errors than Regular or Margin BT. During data collection, we observe that most annotation errors (i.e. proxied by the disagreement between annotator’s indicated preference and the mean aggregated preference across annotators) are between -1: A is slightly better than B and +1: B is slightly better than A. Comparatively, very few annotation errors are between -3 : A is much better than B and +3: B is much better than A (since humans can easily tell apart very good responses from very bad responses). While Regular BT treats both types of preferences as the same, Scaled BT loss function upweights the ‘much better’ samples, which have less annotation errors.
>
> To quantify the observation, we can calculate the loss in a representative scenario where the delta reward (e.g. reward for chosen response - reward for rejected response) is -1 (A is predicted as better than B) while the ground-truth human annotation is either +1 (B is slightly better than A) or +3 (B is much better than A).
>
>
> | **BT Variant** | **Loss at +1** | **Loss at +3** | **Loss Multiplier (+3 / +1)** |
> |----------------|----------------|----------------|---------------------|
> | Regular BT     | 1.3133         | 1.3133         | 1x                  |
> | Margin BT      | 2.1269         | 4.0181         | 1.89x               |
> | Scaled BT      | 1.3133         | 3.9399         | 3x                  |
>
> As we observe, the loss multiplier coming from ‘much better’ samples for Scaled BT (3x) is substantially larger than for both Regular BT (1x) and Margin BT (1.89x). This means that Scaled BT will be less influenced by loss (i.e. gradient updates) due to annotation errors, which is disproportionately found in ‘slightly better’ samples (weighted one-third times compared to ‘much better’ samples).  Note, that the delta reward of -1 is just an illustrative example, but we find that the loss multiplier for Scaled BT is consistently higher than that for either Regular or Margin BT across a range of negative rewards (i.e. when the model predicted and human-annotated preference are in opposite directions,). For instance, when delta reward is -3, the multiplier is 3x for Scaled BT, 1.49x for Margin BT and 1x for Regular BT.
>
> **2. Is Bradley-Terry reward-modeling more robust to annotation errors than SteerLM Regression?**
>
> In the settings that we are interested in (domain-general reward modeling), we are not aware of a good way to approach this question. This is because it’s hard to establish a trustworthy ground-truth label (e.g. a Likert-score of 4 or a preference score of +3 i.e. B>>>A) in order to “calculate/estimate” this assumed error rate. Typically (e.g. [2]), the human annotated sample is assumed to be the ground-truth and a proportion of the labels can be flipped (i.e. +3 i.e. B>>>A is flipped to -3 A >>> B) to investigate the reward model performance. However, this assumption is often wrong as human annotators do get a proportion of annotations wrong as well, so intentionally flipping X% of sample with original Y% of human annotation errors translates to (1-X)*Y + X*(1-Y) errors (ignoring the % for clarity). In the worst case, when Y is high (e.g. 50 or samples are randomly annotated), then the performance is invariant to X. On the other hand, in the best case, if Y is 0, then errors are only accounted for by X.
>
> In real world datasets, Y is hard to estimate and making strong assumptions (e.g. Y = 5 or Y = 25) in simulation hypothetical settings might not correlate with observations in empirical experiments, given the stochastic nature / regularization capabilities of machine learning training. Applying such estimations to Likert scores can be even more complicated since there are many ways/categories in which the errors can be introduced, as the reviewer noted. As such, a fruitful investigation into this question requires a systematic experimental design. We did not do this in our paper as we are mainly interested in introducing HelpSteer2-Preference (a dataset designed to have minimal annotation errors) and making full use of it using appropriate training objectives. Given the time of the rebuttal period, we believe that rushing into this question without a systematic design (e.g. only following the example given by the reviewer), will risk misleading readers into premature conclusions.

---

> ### Author Response · Authors · 2024-11-21
> **Response to reviewer eXue (Part 3 / 3)**
>
> > Has a prompt filter been applied to ensure that prompts from the rewardbench are not included in the training set?
>
>
> We did not apply a prompt filter to ensure prompts in HelpSteer2 were not in the training set. At the time of submission (1 Oct 2024), we were not aware of potential prompt overlap between HelpSteer2 and RewardBench. Post-submission, we were informed by RewardBench maintainer that there were 42 prompts in HelpSteer2-Preference that overlap with RewardBench prompts (see details at [3]). In this setting, overlap means that there is at least one identical 7-gram between a HelpSteer2-Preference prompt and any RewardBench prompt. This definition is a low-precision but high-recall definition of overlap (i.e. over-estimate of actual overlaps), because in many of these 42 cases, the 7-gram overlap is due to a similar template used between the prompts. For instance, both prompts can have a common starting template such as “I want you to act as a novelist. …” but have rather different content after that.
>
> Regardless, this is a negligible portion of both HelpSteer2-Preference (which has 0.42% overlap out of ~10k total prompts) and RewardBench (which has 1.4% overlap out of ~3k total prompts). We believe this small proportion of overlapping prompt is because HelpSteer2-Preference prompts are sourced from ShareGPT, which might have overlapped with the prompt source of constituent datasets of RewardBench as both depend on people voluntarily sharing their ChatGPT conversations.
>
> Out of an abundance of caution, we repeated our training approach for our best model (Scaled BT initialized Helpfulness only, which scored 93.7 on RewardBench Overall, see Table 1) and the resulting model achieves 93.3 Overall RewardBench with 97.5 Chat, 85.3 Chat-Hard 94.3 Safety and 96.1 Reasoning, which is not very different from the original model. Therefore, we are confident that the impact of the extremely limited overlap in prompts is minimal. Please note that even when prompts are overlapping, the model responses are still generated by different models, hence the reward model cannot ‘memorize’ the reward of a response.
>
> > Additionally, how does SteerLM regression, initialized with the Bradley-Terry model, perform?
>
> When we trained a helpfulness-only SteerLM Regression model initialized with Scaled Bradley Terry model, it has 92.2 Reward Bench (Chat 98.9, Chat-Hard 84.9, Safety 91.5, Reasoning 93.7) which is not better than training a Helpfulness only SteerLM Regression model alone with 93.0 RewardBench (Chat 97.2 Chat-Hard 84.2 Safety 94.6 Reasoning 95.8). We believe this is because it is not useful to train a Helpfulness-Only SteerLM Regression model initialized on a Bradley-Terry model. The Bradley-Terry Model does not generate rewards within a narrow range and instead can go between -35 and 5 (based on empirical observation, see Figure 2 of our paper). Hence, initializing the regression model from Bradley-Terry model can lead to extremely large losses initially (as initial predictions are between -35 and 5 while ground truth helpfulness ranges from 0 to 4), which might not be better than randomly initializing the regression head.
>
> > References
>
> [1] https://arxiv.org/abs/2407.21783 The Llama 3 Herd of Models
>
> [2] https://arxiv.org/abs/2311.09641 RLHFPoison: Reward Poisoning Attack for Reinforcement Learning with Human Feedback in Large Language Models
>
> [3] https://huggingface.co/datasets/natolambert/skyworks-rewardbench-contamination

---

### Author Response · Authors · 2024-11-21
**General Comment to All Reviewers**

Dear Reviewers,

We would like to thank everyone for the valuable feedback. We would like to draw the reviewers attention to new results, and re-emphasize the significance of the results we achieve with the REINFORCE model.

**New Results on LMSYS Chatbot Arena**

Due to the strength of our REINFORCE model on automatic benchmarks (i.e. outperforming GPT-4o and Claude 3.5 Sonnet on Arena Hard, AlpacaEval 2 and MT Bench), we evaluated our model in the LMSYS Chatbot Arena. After receiving thousands of double-blind votes comparing this model’s response to the response of other models to user-provided prompts, the model ranks ahead of several frontier models  including Claude 3.5 Sonnet 20240620, GPT-4o 2024-08-06, and Llama 3.1 405B, despite being only a 70B-parameter model.

**The Significance of Our Contribution**

We are glad to see that the reviewers value our methodological work. However, none of the reviewers have explicitly recognized one of our most important contributions: providing an open-source dataset (with an enterprise-friendly license) that allows the community to train aligned models that can match the performance of frontier models on chat benchmarks (both automatic and human-evaluated), starting with a publicly-accessible Llama-3.1-70B-Instruct model. We believe our provided dataset can greatly support the research community to train and study powerful open-source language models. We therefore kindly ask the reviewers to re-evaluate our contribution in light of this.

---

### Author Response · Authors · 2024-11-29
**Follow-up General Comment to All Reviewers**

Dear Reviewers,

Thank you again for your detailed reviews. As the discussion period only has a few days left, we are wondering if you have any further comments or questions for us to consider.

If our rebuttal (adding more experiments, analysis, and writing) helped to alleviate your initial concerns, we would appreciate it if you would consider raising your scores.

---

### Meta-Review · Area_Chair_rRPY · 2024-12-05

**Metareview:**

## **Summary**
This paper introduces the **HelpSteer2-Preference** dataset, which extends the existing HelpSteer2 dataset by including pairwise preference annotations alongside Likert-5 style ratings. This addition allows for a head-to-head comparison of Bradley-Terry (BT) and Regression-style reward modeling paradigms. The authors further propose a novel method that combines insights from both approaches, yielding state-of-the-art (SOTA) results on RewardBench and strong performance in alignment tasks using reinforcement learning with human feedback (RLHF). The dataset and trained models are open-sourced under a permissive CC-BY-4.0 license, promoting reproducibility and practical use.

## **Strengths**
1. **Data Contribution**: The HelpSteer2-Preference dataset addresses an existing gap by providing data that allows for a fair comparison between BT and Regression-style models, a challenge due to the different data collection formats typically used by these paradigms.
2. **Modeling Insights**: The authors conduct a detailed analysis of the three BT variants (Regular, Margin, and Scaled), identifying Scaled BT as the most effective. They also demonstrate that combining BT with Regression-style approaches leads to better performance.
3. **Empirical Results**: The models trained using HelpSteer2-Preference achieve SOTA performance on RewardBench and outperform strong baselines on alignment benchmarks such as Arena Hard.
4. **Open Sourcing**: The (promised) release of the dataset and trained models ensures accessibility and fosters community-driven advancements.

## **Weaknesses**
1. **Incremental Nature of Contributions**: Some reviewers noted that the hybrid approach, while effective, is not fundamentally novel. The dataset augmentation is viewed as a refinement rather than a solid contribution.
2. **Dependence on HelpSteer2**: While the dataset enhances HelpSteer2, the reliance on its structure could limit its generalizability to other domains or applications. However, the authors provided strong justifications for this reliance.
3. **Evaluation Diversity**: The evaluation focuses on alignment tasks and general chat scenarios, which may not fully generalize to more specialized domains like reasoning or code generation.


## **Recommendations**
The paper demonstrates strong empirical results and provides a valuable dataset that will benefit the community. While the contributions are more practical than theoretical, they represent a meaningful step forward in advancing reward modeling and alignment for large language models. The reviewers generally agree on the rigor of the work, the clarity of the presentation, and its significance for practical alignment tasks.

This work is well-executed, and the dataset is likely to have a broad impact, particularly for researchers working on reward modeling and RLHF. While there is room for further insights, the paper's empirical rigor and open-source contributions make it a strong candidate for acceptance.

**Additional Comments On Reviewer Discussion:**

The authors provided rebuttals addressing reviewers’ concerns:
- They showed that HelpSteer2-Preference provides complementary label information and significantly reduces relative errors compared to HelpSteer2 alone.
- Additional experiments confirmed that the dataset leads to stronger alignment models, with substantial improvements in benchmarks like Arena Hard.
- The authors clarified that their focus is on dataset contribution and empirical results rather than algorithmic novelty.
- They justified the choice of evaluation metrics and datasets as aligned with current community standards.

---

### Decision · Program_Chairs · 2025-01-22

Accept (Poster)